# Evidential Uncertainty and Diversity Guided Active Learning for Scene Graph Generation

**Shuzhou Sun[1], Shuaifeng Zhi[2], Janne Heikkilä[1], Li Liu[2,1,†]**
[1]University of Oulu, [2]National University of Defense Technology
[†]Corresponding author: Li Liu {dreamliu2010}@gmail.com

## Abstract

Scene Graph Generation (SGG) has already shown its great potential in various downstream tasks, but it comes at the price of a prohibitively expensive annotation process. To reduce the annotation cost, we propose using Active Learning (AL) for sampling the most informative data. However, directly porting current AL methods to the SGG task poses the following challenges: 1) unreliable uncertainty estimates and 2) data bias problems. To deal with these challenges, we propose EDAL (**E**vidential Uncertainty and **D**iversity Guided Deep **A**ctive **L**earning), a novel AL framework tailored for the SGG task. For challenge 1), we start with Evidential Deep Learning (EDL) coupled with a global relationship mining approach to estimate uncertainty, which can effectively overcome the perturbations of open-set relationships and background-relationships to obtain reliable uncertainty estimates. To address challenge 2), we seek the diversity-based method and design the Context Blocking Module and Image Blocking Module to alleviate context-level bias and image-level bias, respectively. Experiments show that our AL framework can approach the performance of a fully supervised SGG model with only about $10\%$ annotation cost. Furthermore, our ablation studies indicate that introducing AL into the SGG will face many challenges not observed in other vision tasks that are successfully overcome by our new modules.

## 1 Introduction

Scene Graph Generation (SGG) (Johnson et al., 2015) aims at generating a structured representation of a scene that jointly describes objects and their attributes, as well as their pairwise relationships. SGG has attracted significant attention as it provides rich semantic relationships of the visual scenes and has great potential for improving various other vision tasks, such as object detection (Ren et al., 2015; Redmon et al., 2016), image search (Gong et al., 2012; Noh et al., 2017), and visual question answering (Antol et al., 2015; Zhu et al., 2016). Albeit being an emerging area of research, which can bridge the gap between computer vision and natural language processing, SGG is still underexplored despite many recent works focusing on SGG (Chang et al., 2021; Zhu et al., 2022).

The main challenges that impede the advancement of SGG are twofold. On the one hand, existing datasets for SGG (Krishna et al., 2017; Lu et al., 2016) suffer from many serious issues, such as long-tailed distribution, noisy and missing annotations, which makes it difficult to supervise a satisfactory model. On the other hand, existing deep learning-based SGG methods are data hungry, requiring tens or hundreds of labeled samples. However, acquiring high-quality labeled data can be very costly, which is especially the case for SGG. The reason for this is that SGG involves labeling visual *<subject, relationship, object>* triplets (*e.g.*, *<people, ride, bike>*) over entity and relationship classes in an image, which can be difficult and time consuming (Yang et al., 2021; Shi et al., 2021; Guo et al., 2021). Therefore, it is highly desirable to minimize the number of labeled samples needed to train a well-performing model. Active Learning (AL) provides a solid framework to mitigate this problem (Yoo & Kweon, 2019; Kirsch et al., 2019; Huang et al., 2010; 2021). It is, therefore, natural to investigate whether AL can be used to save labeling costs while maintaining accuracy, which is the focus of this paper. In AL, the model selects the most informative examples from an unlabeled pool according to some criteria for manual labeling, and then the model is retrained and evaluated with the selected examples. This looks intuitive yet simple, but directly transferring existing AL methods to the SGG task will face several challenges.

First, existing batch query-based AL paradigms (Gudovskiy et al., 2020; Kim et al., 2021; Mahmood et al., 2021; Sener & Savarese, 2017; Tan et al., 2021) used for SGG face a large number of **open-set relationships**, *i.e.*, relationships that appear in the unlabeled pool but are absent in labeled data, mainly because of the severe long-tailed distribution of the SGG relationships (Zellers et al., 2018; Tang et al., 2020a;b). We observe that existing uncertainty estimation approaches perform badly in classifying SGG relationships, especially open-set relationships. Inspired by Evidential Deep Learning (EDL) (Sensoy et al., 2018) and its advanced performance in open-set action recognition (Bao et al., 2021), we enhance and incorporate it into our proposed AL framework to estimate the relationship uncertainty. Second, the relationship annotations in SGG dataset are very sparse, resulting in severe foreground-background imbalance (Xu et al., 2017; Goel et al., 2022). **Foreground-relationships** are those within annotated triplets in the dataset, while **background-relationships** are the ones that are absent between object pairs. Due to the large number of background-relationships, they can perturb or even dominate the uncertainty estimation. To this end, we propose a relationship mining module, Relationship Proposal Graph (RPG), as a part of the uncertainty estimation, which works by filtering out background-relationships to refine the uncertainty obtained by EDL. Third, despite our improved EDL having the capability to generate reliable estimates of relationship uncertainty, its sampling results are still vulnerable to the problems of traditional uncertainty-based AL methods, *i.e.*, data bias problems (Kim et al., 2021; Shen et al., 2017; Luo et al., 2013). More importantly, we also found that uncertainty-based AL used for SGG will be biased at both context-level and image-level, where the **context** in SGG refers to the feature space formed by relationship triplets. For this issue, we design the Context Blocking Module (CBM) and the Image Blocking Module (IBM), which are inspired by diversity-based AL methods. The former can block similar contexts to avoid the context-level bias, while the latter can block redundant images to eliminate the image-level bias.

**Contributions**. The main contributions of this work are the following: (1) We carry out a pioneering study of using AL for SGG to achieve label efficiency without significantly sacrificing performance loss and propose a novel framework dubbed **E**vidential Uncertainty and **D**iversity Guided Deep **A**ctive **L**earning (EDAL). (2) In the proposed EDAL framework, we introduce novel evidential uncertainty to guide deep active learning and efficient one-shot estimation of relationship uncertainty. In this process, a relationship mining module is designed to avoid the perturbation of uncertainty estimation by background-relationships. In order to effectively mitigate context-level and image-level bias problems induced by AL, we design two modules, CBM and IBM. (3) Extensive experimental results on the SGG benchmarks demonstrate that EDAL can significantly save human annotation costs, approaching the performance of a fully supervised model with only about $10\%$ labeling cost.

## 2 RELATED WORK

**Scene Graph Generation (SGG).** SGG extracts a structured representation of the scene by assigning appropriate relationships to object pairs and enables a more comprehensive understanding of the scene for intelligent agents (Johnson et al., 2015; Lu et al., 2016; Krishna et al., 2017; Liu et al., 2021; Yin et al., 2018). For supervised training of the SGG task, a massive amount of triplets within images in the form of *<subject, relation, object>* need to be provided, which involves several sub-tasks including object detection, object recognition and relationship description, and results in an unaffordable annotation cost. To mitigate this, (Chen et al., 2019) proposed a semi-supervised method for SGG, which requires only a small amount of labeled data for each relationship and generates pseudo-labels for the remaining samples using image-agnostic features. However, these pseudo-labels tend to converge to a few dominant relationships. (Ye & Kovashka, 2021) designed a weak supervision framework to reduce the reliance on labor-intensive annotations with the help of linguistic structures. Recently, (Yao et al., 2021) trained an SGG model in an unsupervised manner by drawing on knowledge bases extracted from the triplets of web-scale image captions. Despite showing the promise of label efficient learning techniques in SGG, the above caption-based methods rely on large-scale external linguistic knowledge which fits the target scene. This, to some extent, limits its generalization to other scenes without adequate linguistic priors. We explore an alternative approach and propose a hybrid AL framework tailored to the SGG task in order to avoid the expensive labeling cost without access to external knowledge.

**Active Learning (AL).** AL aims to select the most informative data from the unlabeled pool for annotation to support model training. In vision tasks such as image classification and object detec-

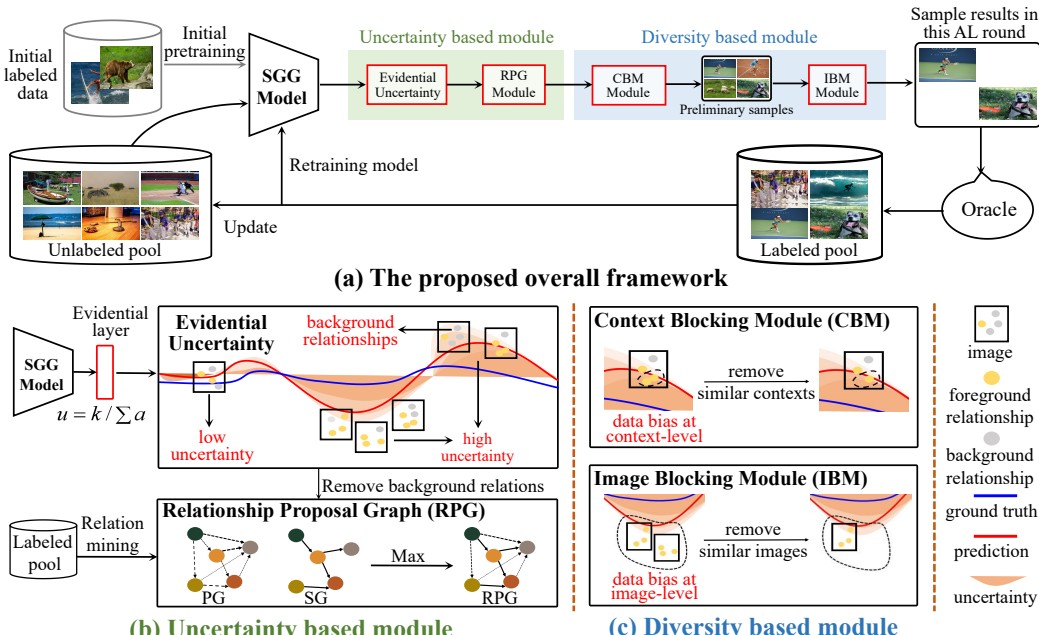

Figure 1: The overall structure of our proposed AL framework EDAL. Following the standard AL setup, EDAL samples data from the unlabeled pool round by round to support the model training, and the quit condition is that the label budget is exhausted.

tion, AL has been widely applied with impressive performance. Uncertainty-based methods (Shen et al., 2017; Luo et al., 2013; Yoo & Kweon, 2019; Huang et al., 2021; Gudovskiy et al., 2020; Kim et al., 2021) and diversity-based methods (Mahmood et al., 2021; Sener & Savarese, 2017; Tan et al., 2021) are currently two types of mainstream AL technologies. For the unlabeled pool, the former aims to estimate uncertainty robustly, while the latter samples the data as dispersedly as possible within their feature space. However, the uncertainty-based methods tend to bias towards a subset of the unlabeled pool, mainly caused by unreliable uncertainty estimation with little labeled data, and the diversity-based methods often struggle with large-scale datasets and complex tasks (Mahmood et al., 2021; Shi et al., 2021). Inspired by the above facts, we propose a hybrid AL strategy with elements from both uncertainty-based and diversity-based methods, which is, to the best of our knowledge, the first hybrid AL method for the challenging SGG task.

**Evidential Deep Learning (EDL).** The pioneering work (Sensoy et al., 2018) proposes using EDL to estimate reliable classification uncertainty, especially for open-set entities, without loss of performance. The authors design a predictive distribution for classification by placing a Dirichlet distribution. Deep Evidential Regression (Amini et al., 2020) introduces the evidential theory to regression tasks by placing evidential priors over the original Gaussian likelihood functions. Recently, (Bao et al., 2021) used EDL to estimate uncertainty for open-set action recognition. A model calibration method is also proposed to regularize EDL training and thus mitigate the overfitting problem. In this paper, we employ EDL to perform uncertainty estimation in the proposed hybrid AL framework. Our main motivation lies in its advantages for reliable open-set uncertainty estimation and the fact that a large number of open-set relationships exist for AL algorithms in the SGG task.

## 3    METHOD

The overall pipeline of EDAL is shown in Figure 1, which is a hybrid AL model composed of uncertainty-based and diversity-based methods. First, the evidential uncertainty estimation method is applied with the extracted prior information from the available labeled data samples to estimate the relationship uncertainty (Section 3.1). Diversity-based sampling driven by context-level and image-level biases is then adopted to refine selected samples to reach the labeling budget (Section 3.2). A pseudo-code of EDAL is given in Appendix A.1.

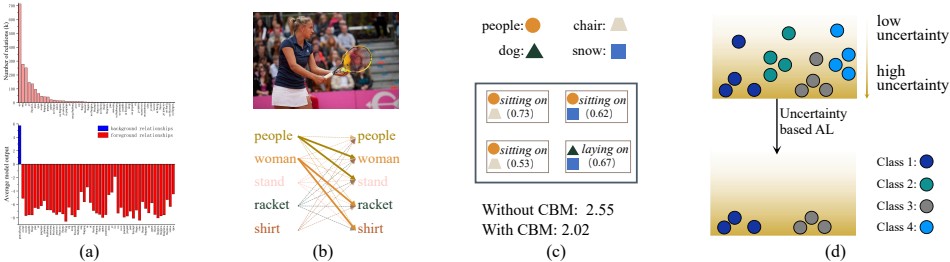

Figure 2: The motivations for key designs in EDAL. (a) motivation for EDL: relationship distributions (top) and average output logits of each relationship (bottom). (b) motivation for RPG: foreground relationship (solid lines) and background relationship (dotted lines). (c) motivation for CBM: the target scene contains many similar contexts (sitting on in this example). (d) motivation for IBM: the sampling results are biased towards partial relationships (Class 1 and 3) in this example.

### 3.1 EVIDENTIAL UNCERTAINTY BASED SAMPLING

Assume that the unlabeled data of the $t-$th AL round is $\mathcal{U}_t = \{I_1, I_2, \cdots I_{|\mathcal{U}_t|}\}$, and that the i-th image $I_i$ contains $|I_i|$ objects and at most $|I_i| \times (|I_i| - 1)$ relationship triplets $\{< o_i^1, \_, o_i^2 >, < o_i^1, \_, o_i^3 >, \cdots, < o_i^{|I_i|}, \_, o_i^{|I_i|-1} >\}$. In Section 3.1.1, we design an uncertainty estimation method for each relationship. Then, in Section 3.1.2, we overcome the perturbation of the background-relationship to the uncertainty estimate by the proposed relationship mining module.

### 3.1.1 EVIDENTIAL UNCERTAINTY ESTIMATION

The SGG task suffers from severe long-tailed distribution of relationships, as shown in Figure 2 (a). In the VG150 dataset(Xu et al., 2017), the top-5 categories amount to more than half, of which we regard top-5 relationships as the head categories and the rest as the tail categories. It is expected that the random samples of the first AL round will only cover a small number of categories and, thus, inevitably yield a lot of open-set relationships in the unlabeled pool. However, commonly adopted probability-based losses (*e.g.*, cross-entropy loss with Softmax probabilities) often provide unreliable uncertainty estimates, such as false over-confidence predictions, for open-set samples (Sensoy et al., 2018). On the contrary, EDL is able to predict reliable uncertainty estimates in this scenario by collecting evidence from each output class by placing a Dirichlet distribution over the class probabilities. For an open-set relationship that does not belong to any available labeled classes, ideally, the model would not be able to collect any evidence that the input belongs to a known relationship. In this case, the parameters of the Dirichlet distribution become all one and lead to an output with a high uncertainty value of 1. Inspired by EDL and its impressive performance in open-set action recognition (Bao et al., 2021), we introduce the modified EDL to estimate the uncertainty of relationships.

Specifically, we denote an object pair and its ground truth relationship as $x_i$ and $y_{ij}$, respectively, where $j \in \{1, 2, \cdots, k\}$ is the index of $k$ relationship categories. Hence $y_{ij} = 1$ if $x_i$ owns the $j$-th relationship, and 0 otherwise. The vanilla EDL supervises the $t$-th AL model $F_t$ with the following loss function to determine the expected probability $p_{ij}$ that $x_i$ belongs to the $j$-th category:

$$L_{EDL}\left(F_t\right) = \underbrace{\sum_{j=1}^{K} \left(\mathrm{y}_{ij} - \hat{\mathrm{p}}_{ij}\right)^2 + \frac{\hat{\mathrm{p}}_{ij}\left(1 - \hat{\mathrm{p}}_{ij}\right)}{\left(S_i + 1\right)}}_{(1)} + \underbrace{KL\left[D\left(\mathbf{p}_i \mid \tilde{\boldsymbol{\alpha}}_i\right) \| D\left(\mathbf{p}_i \mid < 1, \cdots, 1 >\right)\right]}_{(2)}, \quad (1)$$

where $e_{ij}$ is the $j$-th output logit of $x_i$, and it acts as the collected evidence. $\alpha_{ij} = e_{ij} + 1$, $S_i = \sum_{i=1}^{k} \alpha_{ij}$ is the strength of the Dirichlet distribution $D(\mathbf{p_i} \mid \boldsymbol{\alpha_i})$, where $\mathbf{p_i}$ is a simplex representing class assignment probabilities, $\boldsymbol{\alpha_i} = \langle \alpha_{i1}, \ldots, \alpha_{ik} \rangle$. $\hat{\mathrm{p}}_{ij} = \alpha_{ij}/S_i$ is the EDL output for $x_i$. $KL\left[\cdot \| \cdot\right]$ is the KL divergence loss used for regularization, and $\tilde{\alpha}_i$ is the Dirichlet parameters after removal of the non-misleading evidence from predicted parameters $\alpha_i$. During inference, the uncertainty of $x_i$ can be estimated via evidential uncertainty $u_i = k/S_i$ with a maximum value of 1. We refer our readers to (Sensoy et al., 2018) for more details of evidential theory and its derivations. Therefore, for image $I_i$, we can estimate the uncertainty of its triplets and denote the

result as $U_{I_i} = \{U_{<o_i^1,\_,o_i^2>}, U_{<o_i^1,\_,o_i^3>}, \cdots, U_{<o_i^{|I_i|},\_,o_i^{|I_i|-1}>}\}$. However, even though term (2) of Equation 1 has been used as a regularizer, we found that the above vanilla EDL loss suffers from a severe overfitting problem when supervising the SGG model (see Appendix A.2). We argue that the insufficient regularization is due to the large number of background-relationships in the SGG task making the prediction of foreground-relationships indiscriminate, and this intractable problem is shown in 2 (a). Besides, the vanilla EDL loss may be hampered by incidental vanishing gradient because its gradient is $\nabla_{L_{EDL}} = \frac{1}{N}\left[\sum_{i=1}^N \sigma\left(f\left(x_i\right) - y_{ij}\right)\sigma'\left(f\left(x_i\right)\right)\right]$, where $\sigma$ is the ReLU activation function (Sensoy et al., 2018), and therefore, $\sigma' = 0$ when $f\left(x_i\right) < 0$. For the above issues, we find that the **e**vidential-**p**robability based **c**ross-**e**ntropy loss $L_{epce}$ can be used as a strong regularizer when replacing term (2) of Equation 1. Specifically, the final loss of our EDL can be formulated as:

$$L_{EDL}\left(F_t\right) = \underbrace{\sum_{j=1}^K \left(y_{ij} - \hat{p}_{ij}\right)^2 + \frac{\hat{p}_{ij}\left(1 - \hat{p}_{ij}\right)}{\left(S_i + 1\right)}}_{(1)} + \underbrace{\left[-\log\left(\hat{p}_{ij}\right)y_{ij}\right]}_{(2):L_{epce}}. \tag{2}$$

The cross-entropy loss is often paired with softmax, the standard output of classification models, which provides an inflated probability due to its exponential transformation on the output. However, $L_{epce}$ penalizes evidential-probability, which is a second-order probability derived from the parameterized Dirichlet distribution (Sensoy et al., 2018). We argue that $L_{epce}$ can effectively penalize non-discriminative prediction on foreground-relationships, and, thus, can be seen as a strong regularizer. More importantly, cross-entropy loss based on evidential-probability can prevent overconfident predictions in open-set relationships without affecting the uncertainty estimates.

### 3.1.2 RELATIONSHIP PROPOSAL GRAPH

The relationship annotations in the SGG dataset are very sparse and cause a severe foreground-background imbalance problem (Xu et al., 2017; Goel et al., 2022). For example, with five objects in Figure 2 (b), there are, in principle, twenty possible relationships, but only four are annotated and therefore belong to foreground relationships (*e.g.*, *<woman, holding, racket>*, etc.). The large number of background-relationships is dominant and may disturb uncertainty estimation for sparse and valuable foreground ones within the scene. An intuitive idea is to use off-the-shelf methods which can recognize whether a relationship belongs to the foreground or not, such as Directed Semantic Action Graph (Liang et al., 2017), Confusion matrix (Hwang et al., 2018), Statistical prior (Guo et al., 2021), etc. Unfortunately, these methods are all driven by a large amount of labeled data, and the limited amount of annotation in the AL setting is not enough for them. To tackle this issue, we further leverage global information across available labeled images to refine the instance uncertainty obtained in Section 3.1.1. Concretely, a relationship mining method is proposed to infer how likely there exists a foreground-relationship $r$ between two object classes $p$ and $q$ (*i.e.*, instance agnostic), which serves as a weight to refine evidential uncertainty estimation.

First, we generate proximity cluster $\mathcal{C}_{<p,q>} = \{<\mathcal{O}_p, r, \mathcal{O}_q>\}$ to accumulate correlation between two object classes with foreground-relationships from available annotated triplets, where $\mathcal{O}_p$ and $\mathcal{O}_q$ are object instances of classes $p$ and $q$, respectively; $r$ is a foreground-relationship from labeled data. The cluster size $|\mathcal{C}_{<p,q>}|$ is determined by the number of triplets within it. Note that proximity clusters are instance agnostic. Hence triplets from different images can contribute to the same cluster.

We then extract a statistical graph (SG) and a probabilistic graph (PG) from proximity clusters to represent foreground/background probability of a relationship $r$ with different neighboring orders. The statistical graph (SG) models the existence of foreground-relationships whose nodes represent object categories and the edge weights $w_{p,q}^{\text{SG}}$ are binary values depending on whether the corresponding proximity cluster is empty or not. An edge with values of 1 or 0 indicates that the corresponding relationship belongs to the foreground or background, respectively. Considering the lack of representativeness of a few labeled data samples, we further construct a probabilistic graph (PG) which infers the potential likelihood of the relationship $r$ as foreground by taking the second-order neighborhood of the nodes into consideration. Specifically, PG has the same nodes as SG but instead assigns soft edge weights between nodes if they both are connected to a shared neighboring node in SG with non-zero edge weight. By extending proximity cluster to second-order neighboring, we define $\mathcal{C}_{<p,q>}^2 = \{<\mathcal{O}_p, r_1, \mathcal{O}_m>, <\mathcal{O}_m, r_2, \mathcal{O}_q>\}$ where $\mathcal{O}_p, \mathcal{O}_m, \mathcal{O}_q$ are instances of object

classes $p$, $m$ and $q$, respectively; $r_1$ and $r_2$ are two foreground-relationships. We can then compute the edge weight $w_{p,q}^{\mathrm{PG}}$ of PG between classes $p$ and $q$ as follows:

$$w_{p,q}^{\mathrm{PG}} = |\mathcal{C}_{<p,q>}^2|/\mathrm{Max}(1, |\mathcal{C}_{<p,1>}^2|, |\mathcal{C}_{<p,2>}^2|, \cdots, |\mathcal{C}_{<p,\widetilde{k}>}^2|), \quad (3)$$

where $\widetilde{k}$ is the number of object classes. The intuition of PG is that we observe nodes within the second-order foreground neighborhood also possibly correlate via foreground-relationship, and we create PG to take full advantage of such information from the limited labeled data.

Finally, the Relationship Proposal Graph (RPG) is obtained by merging SG and PG to infer our final likelihood of owning foreground-relationship between two classes. RPG has the same nodes as PG and SG whose weights $w_{p,q}^{\mathrm{RPG}}$ are computed via $\mathrm{Max}(w_{p,q}^{\mathrm{PG}}, w_{p,q}^{\mathrm{SG}})$, where $\mathrm{Max}(\cdot, \cdot)$ is an operation kernel taking the maximum edge weights between the same two corresponding nodes. RPG can be thought of as class-wise prior information of the existence of foreground-relationship considering both first and second order adjacent nodes from the pool of labeled triplets. We can use $w_{p,q}^{\mathrm{RPG}}$ to weight $U_{I_i}$ to overcome the perturbation of background-relationships. Specifically, we denote the weighted uncertainty as $U'_{I_i}$, $U'_{I_i} = \{U_{<o_i^1,\_,o_i^2>} \times w_{\widetilde{o}_i^1, \widetilde{o}_i^2}^{\mathrm{RPG}}, U_{<o_i^1,\_,o_i^3>} \times w_{\widetilde{o}_i^1, \widetilde{o}_i^3}^{\mathrm{RPG}}, \cdots, U_{<o_i^{|I_i|},\_,o_i^{|I_i|-1}>} \times w_{\widetilde{o}_i^{|I_i|}, \widetilde{o}_i^{|I_i|-1}}^{\mathrm{RPG}}\}$, where $\widetilde{o}_i^j$ represents the category to which object $o_i^j$ belongs. Please note that as RPG is retrieved using only sampled data during training and its computational overhead is marginal, we, therefore, keep RPG constantly updated whenever new labeled data arrive.

## 3.2 DIVERSITY-BASED BLOCKING MODULE

In this subsection, we introduce the Context Blocking Module (CBM) and the Image Blocking Module (IBM) to cope with the data bias issue of the uncertainty-based AL methods.

As to the context-level data bias illustrated in Figure 2 (c), a scene may contain many similar contexts (*e.g.*, *<people, sitting on, chair>*). We argue that when considering the potential similarity of objects in the learned feature space, not all triplets sharing similar contexts contribute equally to the performance of SGG but instead increase the labeling cost and computational expense. Image-level data bias is also shown in Figure 2 (d). The uncertainty-based sampling will be biased towards the classes where the current model performs poorly (the sampling results will bias toward data from Class 1 and Class 3 in this case), which has been discussed in the existing literature as well (Huang et al., 2021; Mahmood et al., 2021; Sener & Savarese, 2017; Tan et al., 2021). See Appendix A.3 for further discussion of the data bias problem.

To alleviate the data bias problem at the context-level, we propose CBM to block certain triplets/object pairs sharing similar contexts. Let the context feature of $I_i$ in the $t$-th AL round be $f_{I_i,t}$, $f_{I_i,t} = \{f_{I_i,t}^1, f_{I_i,t}^2, \cdots, f_{I_i,t}^{|I_i| \times (|I_i|-1)}\}$, where $f_{I_i,t}^j$ is the $j$-th context feature extracted by the model $F_{t-1}$. We then calculate the context-level density of $I_i$ denoted by $D_{I_i,C} = \{D_{I_i,C}^1, D_{I_i,C}^2, \cdots, D_{I_i,C}^{|I_i| \times (|I_i|-1)}\}$, where $D_{I_i,C}^i = (\sum_{j=1}^{|I| \times (|I_i|-1)} |f_{I_i,t}^i - f_{I_i,t}^j|)^{-1}$. Similar contexts with the largest density value are removed, whose actual number is determined by a pre-set deduplication ratio $\tau$, and we update the uncertainty.

For the data bias problem at the image-level, we propose IBM to filter images that contribute similarly to SGG. Let the labeling budget at $t$-th round is $\ell_t$. Although typical AL methods often directly select $\ell_t$ samples with the largest uncertainty, in EDAL $\widetilde{\ell}_t$ ($\widetilde{\ell}_t > \ell_t$) preliminary samples with the highest uncertainties are first chosen, i.e. $\mathcal{S}_{t,\widetilde{\ell}_t} = \{I_1, I_2, \cdots, I_{\widetilde{\ell}_t}\}$. Similar to the density computation in CBM, we also calculate the image-level density $D_{\mathcal{S}_{t,\widetilde{\ell}_t}} = \{D_{I_1}, D_{I_2}, \cdots, D_{I_{\widetilde{\ell}_t}}\}$, where $D_{I_i,I} = (\sum_{j=1}^{\widetilde{\ell}_t} |f_{I_i,t}^i - f_{I_i,t}^j|)^{-1}$, $f_{I_i,t}^i$ is the $i$-th image feature extracted by the model $F_{t-1}$. Finally, we remove $\widetilde{\ell}_t - \ell_t$ images with the largest image-level density value to reach the desired labeling budget $\ell_t$. Note that because $\ell_t$ in IBM is the actual labeling budget and $\widetilde{\ell}_t$ is a hyperparameter related to the degree of image-level bias. We argue that $\widetilde{\ell}_t$ should be gradually increased during training, as several existing AL works have shown that the data bias problem will become more and more serious as AL progresses. Specifically, $\widetilde{\ell}_t$ in our framework can be computed as $\widetilde{\ell}_t = \ell_t + \lambda |\mathcal{U}_t| t^\eta$, and we call it De-bias Temperature Units (DeTu). In the Appendix A.4, we explain why $\widetilde{\ell}_t$ is modeled in an exponential form rather than constant $\widetilde{\ell}_t = \ell_t + \lambda_1 |\mathcal{U}_t|$ (De-bias

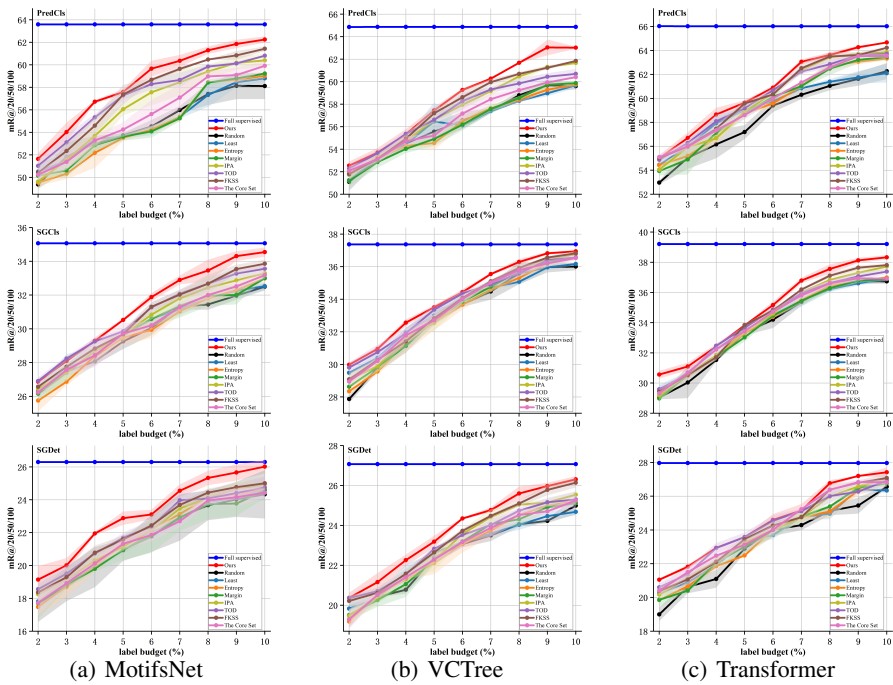

| (a) MotifsNet | (b) VCTree | (c) Transformer |

Figure 3: Active Learning performance of EDAL and baseline methods. mR@/20/50/100 represents the average performance of R@20, R@50, and R@100. We repeat each experiment three times and report the mean (solid line) and standard deviation (shadow).

Constant value, DeCv) or linear form $\widetilde{\ell}_t = \ell_t + \lambda_2 |\mathcal{U}_t| t$ (De-bias Linear Units, DeLu). We will also quantify the advantages of DeTu through the ablation study (Section 4.3).

# 4 EXPERIMENTS

## 4.1 EXPERIMENT SETUP

**Implementations.** We show the effectiveness of EDAL on the popular VG150 dataset (Xu et al., 2017) for the SGG task through extensive experiments. To prepare the AL dataset, we first remove all the relationship annotations from the official training set (about 56k images) to create the initial unlabeled pool $\mathcal{U}$ with object annotations only. We randomly sample from $\mathcal{U}$ at the rate of $p_1$ in the first round and start the AL framework. For the following $i$-th round ($i \geq 2$), sampling with the rate of $p_i$ by the proposed method is used to obtain $\mathcal{S}_{i,\ell_i}$ until the label budget is exhausted. Specifically, in our experiments, $p_1$=0.01, $p_i$=0.01, $i = 2, 3, \cdots$. We set the label budget to be 10%, *i.e.*, to label 10% of the data in $\mathcal{U}$. For $\tau$ in CBM and $\widetilde{\ell}_t = \ell_t + \lambda |\mathcal{U}_t| t^\eta$ in IBM, we set $\tau = 0.2$, $\lambda = 10^{-5}$ and $\eta = 4$. See Appendix B.1 for how these parameters are determined.

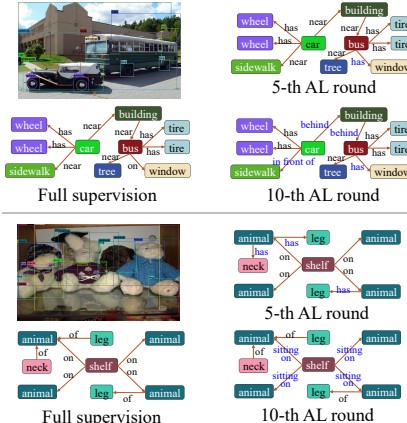

Figure 4: Qualitative examples of the fully supervised model and our method at the 5-th and the 10-th AL rounds. Differences are highlighted in blue.

**Baselines and backbones.** We compare our proposed method against the following baseline methods: **1) Random sampling**. **2) Classical uncertainty-based methods**: including Least Confidence Sampling (LCS) (Shen et al., 2017), Maximum Entropy Sampling (MES) (Luo et al., 2013), and Margin Sampling (MS) (Culotta & McCallum, 2005). **3) Recent state-of-the-art methods**: The Core-set approach (Sener & Savarese, 2017), Fisher Kernel Self Supervision (FKSS) (Gudovskiy

et al., 2020), Integer Programming Approach (IPA) (Mahmood et al., 2021), Loss Prediction Module (LPM) Yoo & Kweon (2019) and Temporal Output Discrepancy (TOD) (Shi et al., 2021). Please refer to Appendix B.2 for implementation details. We use three typical SGG backbones in our evaluation: 1) Neural Motifs backbone (MotifsNet) (Zellers et al., 2018). 2) VCTree (Tang et al., 2020b). 3) Transformer-based backbone (Shi & Tang, 2020).

**Evaluation metrics and modes.** The first is evaluation metrics for AL. We follow the standard evaluation metric, *i.e.*, performance under fixed labeling budget. Under this metric, the higher the performance, the better the AL framework. Then comes the evaluation modes for SGG. Following MotifsNet (Zellers et al., 2018), we use three evaluation modes: 1) Predicate classification (PredCls). 2) Scene Graph Classification (SGCls), and 3) Scene Graph Detection (SGDet).

Table 1: Ablation study results. The SGG backbone used here is MotifsNet.

| Method | PredCls R@/20/50/100 | SGCls R@/20/50/100 | SGDet R@/20/50/100 |
|---|---|---|---|
| EDAL-EDL | 56.0/61.5/63.2 | 29.0/32.0/32.9 | 18.7/24.9/28.8 |
| EDAL-RPG | 55.5/61.9/63.7 | 29.3/32.1/33.0 | 19.0/25.4/29.3 |
| EDAL-IBM | 56.7/62.2/63.8 | 29.9/31.7/32.9 | 18.4/24.4/28.9 |
| EDAL-CBM | 56.8/62.6/64.3 | 30.3/32.8/33.6 | 18.7/25.2/29.3 |
| EDAL | 57.0/63.5/65.1 | 31.7/35.0/35.8 | 21.4/27.2/30.3 |

Table 2: Performance on open-set relationship recognition. The SGG backbone used here is MotifsNet.

| Method | PredCls R@/20/50/100 | SGCls R@/20/50/100 | SGDet R@/20/50/100 |
|---|---|---|---|
| EDAL+LCS | 50.1/62.4/70.1 | 34.3/40.4/43.7 | 14.4/20.4/25.5 |
| EDAL+TOD | 53.2/66.0/73.8 | 34.0/42.2/44.3 | 15.8/22.0/27.0 |
| EDAL+FKSS | 57.3/69.4/76.8 | 35.2/43.1/46.2 | 17.1/23.4/28.8 |
| EDAL+LPM | 55.2/68.4/75.6 | 34.7/42.6/45.3 | 16.3/22.7/28.1 |
| EDAL | 59.7/72.7/80.6 | 37.1/44.6/48.4 | 17.7/25.2/30.3 |

## 4.2 MAIN RESULTS AND ANALYSIS

In this section, we compare EDAL with other baseline methods. We report the quantitative results in Figure 3 and show some qualitative samples in Figure 4. From the above results, we have several observations: **1)** Our proposed AL framework shows improvement with a clear margin over all baseline methods, *i.e.*, EDAL has higher performance under fixed label budgets. **2)** The standard deviation (shadows in Figure 3) shows that our proposed AL framework has a more stable learning process. We believe that this mainly comes from CBM and IBM, which can alleviate the data bias problem at the image and context-level to ensure that the results of each AL

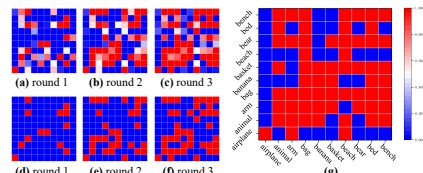

(a) round 1 (b) round 2 (c) round 3

(d) round 1 (e) round 2 (f) round 3 (g)

Figure 5: Relationship heatmaps obtained by RPG (a, b, c), simple binary statistics (d, e, f), and the full training set (g). Here we only show 10 object categories for clarity.

round can cover the unlabeled pool better, while the biased data obtained by baseline strategies can obviously mislead the model training. **3)** The uncertainty-based methods (*e.g.*, TOD (Huang et al., 2021), FKSS (Gudovskiy et al., 2020), LCS (Shen et al., 2017), etc.) perform better in the first few AL rounds but lag in the subsequent rounds, and we conjecture that the data bias problem is responsible for this, which gets worse as AL progresses. This phenomenon also justifies our claim in section 3 that the hyperparameter $\widetilde{\ell}_t$ in IBM to alleviate data bias should be gradually increased. Thus, we argue that the exponential deduplication strategy DeTu is key to our proposed AL framework. **4)** The diversity-based methods (*e.g.*, IPA (Mahmood et al., 2021), the Core-set approach (Sener & Savarese, 2017), etc.) perform very poorly due to the large dataset in the SGG task. However, our hybrid AL framework can avoid the curse of scale by leveraging uncertainty to obtain preliminary samples. **5)** Figure 4 shows that our method can output more fine-grained/meaningful relationships, *e.g.*, *<animal, sitting on, shelf> VS <animal, on, shelf>*. We argue that this improvement stems from our uncertainty estimation method giving more attention to tail categories, which we discuss in more detail in Appendix B.3. **6)** Due to space limitations, we only report average performance mR@/20/50/100 here instead of individual R@/20, R@/50 and R@/100. For the detailed results, see Appendix B.4. Furthermore, Figure 3 shows the results under the most commonly used metrics for AL, we also report the performance under another metric, labeling budget under expected model performance, in Appendix B.5. Finally, the label budget set in this paper is 10%, but we also explored the model performance under more budgets with the results presented in Appendix B.6.

## 4.3 ABLATION STUDY

As discussed in Section 3 and illustrated in Figure 1, our proposed AL framework consists of four key designs, *i.e.*, EDL-based uncertainty estimation module, RPG, CBM, and IBM. We

drop the above designs one by one from our framework and report the numbers in Table 1. Note that to keep the framework running, the removed EDL-based module will be replaced by a baseline uncertainty estimation method (we use LCS (Shen et al., 2017) here). The above quantitative results show that each design can contribute to our AL framework. In the following, we will further prove the effectiveness of those designs and explore their functionality.

**Ablation on EDL-based uncertainty estimation module.** This subsection evaluates the robustness of our proposed uncertainty method. To evaluate this advantage, we report open-set recognition performances of different uncertainty methods in Table 2. Our evaluation procedure is inspired by (Kopetzki et al., 2021), which emphasizes that a robust uncertainty method should perform well for out-of-distribution (OOD) data. The results show that the EDL-based module outperforms other methods by a clear margin. We argue these improvements are achieved by placing a Dirichlet distribution on the class probabilities to avoid over-confident predictions on the open-set relationships. Due to space constraints, we give the implementation details in Appendix B.7.

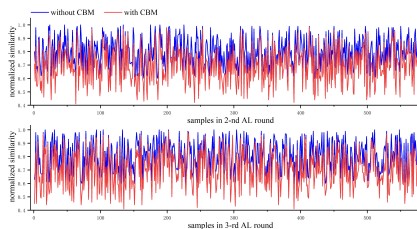

Figure 6: Normalized similarity of samples at different AL rounds.

**Ablation on RPG.** As shown in Figure 5, simple binary statistics can only cover a very small number of relationships, especially in the early AL rounds. Therefore, we argue that in the AL setting, simple binary statistics cannot complete the task of filtering the background-relationships. In contrast, our method leverages the second-order neighboring to complement the binary statistics. We can find that even in the early AL rounds (*e.g.*, round 1 (Figure 5 (a)), RPG can still obtain satisfactory relationship mining results, and in the third round, our method can obtain mining results that cover almost all the true relationships in the training set.

**Ablation on CMB**. Figure 6 shows the average similarity of the context features in each sample, which illustrates that CBM can remove duplicate information and provide

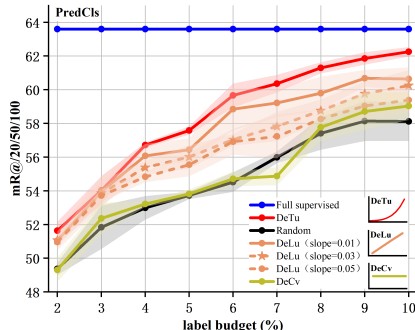

Figure 7: Performance of different deduplication strategies in IBM. The SGG backbone used here is MotifsNet.

a more meaningful uncertainty estimate. Furthermore, we find that different AL rounds of CBM (with the same $\tau$) have consistent deduplication effects. This is expected as we believe that the context-level bias is only related to the data itself, which is why we set $\tau$ to a constant value.

**Ablation on IBM**. We show different deduplication strategies in Figure 7, and we have the following observations. DeCv is a lot behind DeTu and DeLu, which fully supports our claim in Section 3 that different AL rounds have different uncertainty estimation capabilities and should use a progressively improved deduplication strategy. At the same time, in the early AL rounds (*e.g.*, 2-5 rounds), DeLu can be on par with DeTu, but in the later rounds, their gap widens. We argue that this reflects the fact that the linear form is no longer sufficient to keep pace with the degree of the data bias problem due to the improvement of uncertainty estimation ability.

## 5 CONCLUSION

Compared with simple tasks, AL for SGG suffers from a large number of complex open-set relationships, background-relationships that perturb the uncertainty estimation, as well as data bias problems at both the image-level and context-level. For the above challenges, in this work, we have developed a hybrid Active Learning framework, EDAL, for releasing the expensive relationship annotation cost in the SGG task. We have extensively experimented on different types of SGG backbones, and the results in three SGG evaluation modes show that our proposed AL framework outperforms other state-of-the-art methods by a clear margin. EDAL consists of four key components, the EDL-based uncertainty estimation module, RPG, CBM, and IBM, and we have also explained their motivations in detail and explored their contributions through the ablation study. Finally, we discuss the limitations of existing metrics in Appendix B.4 and explore the potential of multimodal annotation-based active learning for the SGG task in Appendix C, which belongs to our future work.

ACKNOWLEDGMENTS

This work was partially supported by National Key Research and Development Program of China No. 2021YFB3100800, the Academy of Finland under grant 331883, Infotech Project FRAGES, the National Natural Science Foundation of China under Grant 62022091 and 62201588, the National Natural Science Foundation of China under Grant 62201603, and Research Program of National University of Defense Technology under Grant ZK22-04.

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

# A APPENDIX TO METHOD

## A.1 PESUDO-CODE FOR THE EDAL

Without loss of generality, we show the sampling process for $t-$th AL round ($t \geq 2$). Assume that the unlabeled data of the $t-$th AL round is $\mathcal{U}_t = \{I_1, I_2, \cdots I_{|\mathcal{U}_t|}\}$, and that the i-th image $I_i$ contain $|I_i|$ objects and thus $|I_i| \times (|I_i| - 1)$ relationship triplets $\{< o_i^1, \_, o_i^2 >, < o_i^1, \_, o_i^3 >, \cdots, < o_i^{|I_i|}, \_, o_i^{|I_i|-1} >\}$.

---

**Algorithm 1** Evidential Uncertainty and Diversity Guided Deep Active Learning (**EDAL**)

---

**Input:** $t, \mathcal{U}_t = \{I_1, I_2, \cdots I_{|\mathcal{U}_t|}\}, \tau, \ell_t, \lambda, \eta$     // inputs of $t-$th AL round
**Output:** $\mathcal{S}_{t, \ell_t}$

1: **for** $i = 1$ to $|\mathcal{U}_t|$ **do**
2:     $U_{I_i} = \{U_{<o_i^1, \_, o_i^2>}, U_{<o_i^1, \_, o_i^3>}, \cdots, U_{<o_i^{|I_i|}, \_, o_i^{|I_i|-1}>}\}$     // estimate the uncertainty of each triplet using evidential uncertainty estimation (Section 3.1.1)
3:     $w_{I_i}^{\text{RPG}} = \{w_{\tilde{o}_i^1, \tilde{o}_i^2}^{\text{RPG}}, w_{\tilde{o}_i^1, \tilde{o}_i^3}^{\text{RPG}}, \cdots, w_{\tilde{o}_i^{|I_i|}, \tilde{o}_i^{|I_i|-1}}^{\text{RPG}}\}$     // calculate the RPG weight for each triple in $I_i$
4:     $U_{I_i}' = \{U_{<o_i^1, \_, o_i^2>} \times w_{\tilde{o}_i^1, \tilde{o}_i^2}^{\text{RPG}}, U_{<o_i^1, \_, o_i^3>} \times w_{\tilde{o}_i^1, \tilde{o}_i^3}^{\text{RPG}}, \cdots, U_{<o_i^{|I_i|}, \_, o_i^{|I_i|-1}>} \times w_{\tilde{o}_i^{|I_i|}, \tilde{o}_i^{|I_i|-1}}^{\text{RPG}}\}$
   // refine the evidential uncertainty (Section 3.1.2)
5: **end for**
6: $U_{\mathcal{U}_t} = \{U_{I_1}', U_{I_2}', \cdots, U_{I_{|\mathcal{U}_t|}}'\}$     // reliable uncertainty obtained by Section 3.1
7: **for** $i = 1$ to $|\mathcal{U}_t|$ **do**
8:     $D_{I_i, C} = \{D_{I_i, C}^{<o_i^1, \_, o_i^2>}, D_{I_i, C}^{<o_i^1, \_, o_i^3>}, \cdots, D_{I_i, C}^{<o_i^{|I_i|}, \_, o_i^{|I_i|-1}>}\}$     // calculate the density at context-level (Section 3.2)
9:     remove the $\lceil |I_i| \times (|I_i| - 1) \times \tau \rceil$ context with the smallest density, denoted as $D_{I_i, C}'$, $D_{I_i, C}' = \{D_{I_i, C}^1, D_{I_i, C}^2, \cdots, D_{I_i, C}^{|D_{I_i, C}'|}\}$     // alleviate the context-level bias
10:    get the uncertainty of each context in the $D_{I_i, C}'$ according to $U_{\mathcal{U}_t}$, denoted as $U_{I_i, C}$, $U_{I_i, C} = \{U_{I_i, C}^1, U_{I_i, C}^2, \cdots, U_{I_i, C}^{|D_{I_i, C}'|}\}$
11:    get the uncertainty of $I_i$, $U_{I_i}'' = \sum_{j=1}^{|D_{I_i, C}'|} U_{I_i, C}^j / |D_{I_i, C}'|$
12: **end for**
13: $U_{\mathcal{U}_t}' = \{U_{I_1}'', U_{I_2}'', \cdots, U_{I_{|\mathcal{U}_t|}}''\}$     // uncertainty obtained by Context Blocking Module 3.2. Compared with $U_{\mathcal{U}_t}$, $U_{\mathcal{U}_t}'$ does not contain context-level bias
14: $\widetilde{\ell}_t = \ell_t + \lambda |\mathcal{U}_t| t^\eta$
15: select $\widetilde{\ell}_t$ samples with the highest uncertainty, denoted as $\mathcal{S}_{t, \widetilde{\ell}_t}$, $\mathcal{S}_{t, \widetilde{\ell}_t} = \{I_1, I_2, \cdots, I_{\widetilde{\ell}_t}\}$
16: **for** $i = 1$ to $\widetilde{\ell}_t$ **do**
17:    $D_{I_i, I} = \{D_{I_i, I}^1, D_{I_i, I}^2, \cdots, D_{I_i, I}^{\widetilde{\ell}_t}\}$     // calculate the density at image-level (Section 3.2)
18: **end for**
19: $D_{\mathcal{S}_{t, \widetilde{\ell}_t}} = \{D_{I_1}, D_{I_2}, \cdots, D_{I_{\widetilde{\ell}_t}}\}$
20: remove the $\widetilde{\ell}_t - \ell_t$ samples with the smallest density, denoted as $\mathcal{S}_{t, \ell_t}$     // alleviate the image-level bias. $\mathcal{S}_{t, \ell_t}$ is the final sampling result of $t-$th AL round

---

## A.2 LOSS FUNCTION

As shown in Table 3, the vanilla EDL loss function performs very poorly on the SGG task. We think this is mainly because the regularization term (2) of Equation 1 is insufficient. We prove it by reporting the performance of Equation 1 without term (2), and the results validate our point that term (2) can only bring about trivial improvement. Instead, our loss function can provide the model with better supervision without compromising the estimation of uncertainty. We have discussed this fully in Section 3.

Table 3: Model performance with different loss functions.

| method | PredCls R@20/50/100 | SGCls R@20/50/100 | SGDet R@20/50/100 |
|---|---|---|---|
| term (1) in Equation 1 | 56.1/60.7/62.1 | 28.7/31.3/32.4 | 17.3/23.5/27.5 |
| Equation 1 | 55.4/60.9/62.6 | 29.5/34.9/32.5 | 18.6/24.1/27.5 |
| Equation 2 (ours) | 57.0/63.5/65.1 | 31.7/35.0/35.8 | 21.4/27.2/30.3 |

### A.3 DATA BIAS PROBLEM IN ACTIVE LEARNING

Active learning, especially those based on uncertainty estimation, is prone to data bias problems. This has been discussed in much of the existing literature (Huang et al., 2021; Mahmood et al., 2021; Sener & Savarese, 2017; Tan et al., 2021). Here we further explain how active learning produces biased sampling.

Let $Q_{(x,y)}$ and $P_{(x,y)}$ denote the distribution of the unlabeled data pool and the selected data obtained by an AL method, and suppose their densities are $q(x,y) = q(y \mid x)q(x)$ and $p(x,y) = p(y \mid x)p(x)$, respectively. We use $\mathcal{H}(h \sim H)$ to represent the optimal sampling for the original distribution $H$ under the condition of a given sampling rate, where $h$ obeys the distribution $H$. Based on this definition, $\mathcal{H}((x,y) \sim P_{(x,y)})$ can be calculated as:

$$\mathcal{H}((x,y) \sim Q_{(x,y)}) = -\iint q(y \mid x)q(x) \ln(q(y \mid x)q(x)) d_x d_y . \tag{4}$$

$$\mathcal{H}((x,y) \sim P_{(x,y)}) = -\iint q(y \mid x)q(x) \ln(p(y \mid x)p(x)) d_x d_y . \tag{5}$$

We then use KL divergence $D_{KL}(Q_{(x,y)} \mid\mid P_{(x,y)})$ to describe the extent to which $P_{(x,y)}$ covers $Q_{(x,y)}$:

$$
\begin{aligned}
D_{KL}(Q_{(x,y)} \mid\mid P_{(x,y)}) &= \mathcal{H}((x,y) \sim P_{(x,y)}) - \mathcal{H}((x,y) \sim Q_{(x,y)}) \\
&= \iint q(y \mid x)q(x) \ln \frac{q(y \mid x)q(x)}{p(y \mid x)p(x)} d_x d_y.
\end{aligned}
\tag{6}
$$

Therefore, we can obtain the optimal active learning query function $\mathcal{Q}_{AL}$ by minimizing $D_{KL}(Q_{(x,y)} \mid\mid P_{(x,y)})$:

$$\mathcal{Q}_{AL} = \arg\min_{P_{(x,y)}} D_{KL}(Q_{(x,y)} \mid\mid P_{(x,y)}) . \tag{7}$$

However, from an example shown in Figure 2 (d), we can see that $P_{(x,y)}$ is biased towards partial categories in practice. Assuming that the optimal sampling of training data under given conditions $Q_{\widetilde{AL}}$. Obviously, in $P_{(x,y)}$, some high uncertainty data $Q_{\widetilde{AL}} \backslash \mathcal{H}((x,y) \sim P_{(x,y)})$ are not queried by $\mathcal{Q}_{AL}$, but some low uncertainty data $\mathcal{H}((x,y) \sim Q_{(x,y)}) \backslash Q_{\widetilde{AL}}$ are selected instead. This unreasonable sampling is exactly the data bias problem in active learning.

### A.4 DE-BIAS UNIT

In the Image Blocking Module (IBM), an exponential formulation of De-bias Temperature Units (DeTu) $\widetilde{\ell}_t = \ell_t + \lambda \mid\mathcal{U}_t\mid t^\eta$ is adopted rather than its counterparts of constant form (De-bias Constant value, DeCv) $\widetilde{\ell}_t = \ell_t + \lambda_1 \mid\mathcal{U}_t\mid$ or linear form (De-bias Linear Units, DeLu) $\widetilde{\ell}_t = \ell_t + \lambda_2 \mid\mathcal{U}_t\mid t$ (see Section 3.2).

In addition to the quantitative result from Figure 7, here we further explain the intuition. The uncertainty-based methods suppose that data with higher uncertainty contains more valuable information, while the diversity-based methods believe that data with lower density obtain richer information. Based on these two viewpoints, we define the information of the data $\mathbf{X}$ as $E(\mathbf{X})$ to quantify the value of samples, $E(\mathbf{X}) = U_\mathbf{X}/(D_\mathbf{X} + \delta)$, where $U_\mathbf{X}$ and $D_\mathbf{X}$ are the uncertainty and

density of $\mathbf{X}$, respectively and $\delta$ is an infinitesimal value to avoid the numerical issue. Thus, the information-gain of IBM is:

$$E(\mathcal{S}'_{t,\ell_t}) - E(\mathcal{S}_{t,\ell_t}) = \frac{(D_{\mathcal{S}_{t,\ell_t}} + \delta)U_{\mathcal{S}'_{t,\ell_t}} - (D_{\mathcal{S}'_{t,\ell_t}} + \delta)U_{\mathcal{S}_{t,\ell_t}}}{(D_{\mathcal{S}'_{t,\ell_t}} + \delta) \times (D_{S_{\ell_t}} + \delta)}, \tag{8}$$

where $\mathcal{S}'_{t,\ell_t}$ is the data sampled without IBM, and $\mathcal{S}_{t,\ell_t}$ is the results obtained by our proposed AL framework with IBM. We assume two extreme but representative scenarios to support our choice of DeTu. (1) Assume the current model is uncertainty-unreliable, *i.e.*, it cannot provide any valid uncertainty estimation. In this case, the result of the uncertainty method is equivalent to random sampling. In this case, the IBM will be indistinctive or even play a negative role if *i.e.*, $E(\mathcal{S}'_{t,\ell_t}) - E(\mathcal{S}_{t,\ell_t}) \leq 0$. (2) Assume the current model is uncertainty-reliable, *i.e.*, it provides reasonable uncertainty estimations with the confidence of 1. In this case, considering the fact that uncertainty-based methods suffer from data bias problems and $E(\mathcal{S}'_{t,\ell_t}) - E(\mathcal{S}_{t,\ell_t}) \geq 0$, this indicates that the IBM will always bring a positive effect.

The above facts suggest that different blocking strategies should be used in different AL rounds because the uncertainty estimation ability of the model is gradually improved, so the constant strategy (DeCv) can not meet our requirements. At the same time, for the linear strategy (DeLu), we argue that the DeLu with a large slope may have an adverse impact in the early AL rounds due to the unreliable uncertainty estimation, while the DeLu with a small slope may not meet the demand of data removal in the later AL rounds due to serious data bias problem. Therefore, we believe that IBM needs a nonlinear data removal strategy, and the results in Figure 7 can also prove our point.

## B  APPENDIX TO EXPERIMENT

### B.1  HYPERPARAMETERS SETTING

For $\eta$ and $\lambda$ in IBM. Motivated by the fact that the data bias problem will become more serious as AL progresses, we argue and experimentally prove that the deduplication ratio should be gradually increased during training. The exponential deduplication function is a natural choice that meets the above expectations. The value of $\lambda$ is set according to $\eta$, and the basic principle is to ensure a suitable deduplication amount. In our paper, we keep the deduplication amount at approximately less than 10% via $\lambda$. We show the exponential functions with different $\eta$ and $\lambda$ in Figure 8. We argue that the $\eta$ should be nearly flat in the early AL rounds and should grow rapidly in the later rounds to combat the increasingly serious data bias problem. As a tradeoff, we set $\eta = 4$ and thus $\lambda = 10^{-5}$.

However, for $\tau$ in CBM, we argue that the data bias problem at context-level is only related to the input data itself and thus should be set as a constant value. In Figure 9, we show model performance with different $\tau$. Inspired by this experiment, we set $\tau = 0.2$ in this paper.

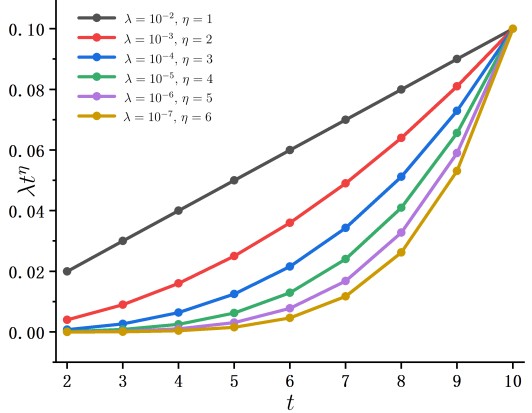

Figure 8: Deduplication function with different $\eta$ and $\lambda$.

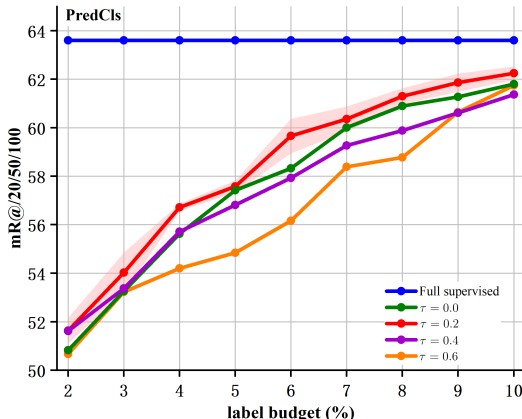

Figure 9: Model performance with different $\tau$.

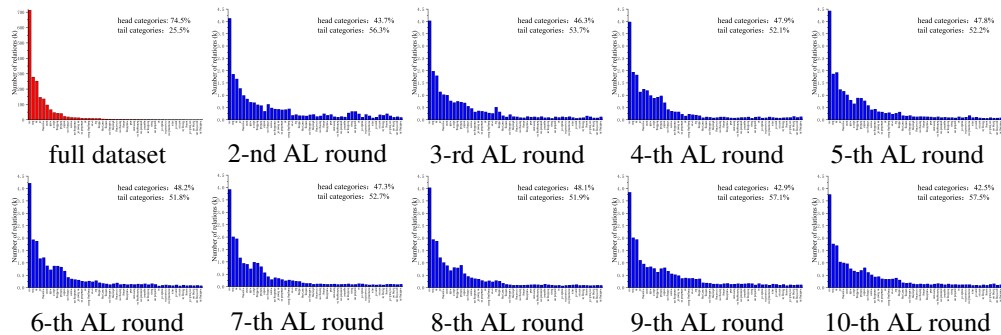

Figure 10: The relationship distributions of full dataset (trainset of VG150) and sampling results of our AL framework.

## B.2 COMPARISON DETAILS OF ACTIVE LEARNING BASELINES

To validate our proposed method, we compare a large number of baseline active learning methods, including classical uncertainty-based methods and recent state-of-the-art methods. Specifically, the comparison details of these baselines are as follows. The Core-set approach (Sener & Savarese, 2017) is a diversity-based AL method, and we follow the training tricks and hyperparameters in the original paper. Fisher Kernel Self Supervision (FKSS) (Gudovskiy et al., 2020) proposes a low-complexity feature density matching method and calculates the uncertainty of unlabeled data based on it. Note, for fairness, we use the complete unlabeled pool when comparing this method rather than using only part of the pool in the original paper to create artificial data. Integer Programming Approach (IPA) (Mahmood et al., 2021) minimizes the discrete Wasserstein distance in feature space from the unlabeled pool to select the core set. Temporal Output Discrepancy (TOD) (Shi et al., 2021) calculates the uncertainty on unlabeled data based on losses at different training stages, and it also incorporates the semi-supervised method to improve the performance. To keep focus and follow the AL setup, we only compare the uncertainty method proposed in TOD.

## B.3 ANALYSIS ON LONG-TAILED DISTRIBUTION

Our proposed EDL-based method can provide reliable uncertainty estimates, and in this subsection we show that it also leads to an important product, i.e., alleviating the long-tail distribution problem. Our uncertainty estimation method pays more attention to tail categories due to these relationships and performs poorly in the head categories-dominated model. Figure 10 shows the relationship distributions of the full dataset and sampling results of our AL framework. We can see that the proportion of head categories in the full data is 78%. However, that in our method does not exceed 50%. We think this improvement is very important for the SGG task because it encourages the model

to predict more fine-grained/meaningful relationships, e.g., predict *sitting on* instead of *on*. Besides, this advantage has the potential to make our method approach or even exceed fully supervised model performance with less label cost, which we have analyzed in Appendix B.6.

Although our method can alleviate the long-tailed distribution problem, it is worth noting that the head categories in SGG are often common relationships that will appear in various scenes. Therefore, although our method gives priority to tail categories, many head category relationships (*on*, *has*, *in*, etc.) that appear in tail category scenarios are also sampled. As a result, we believe our method will perform better in tasks with a low fusion of head and tail categories.

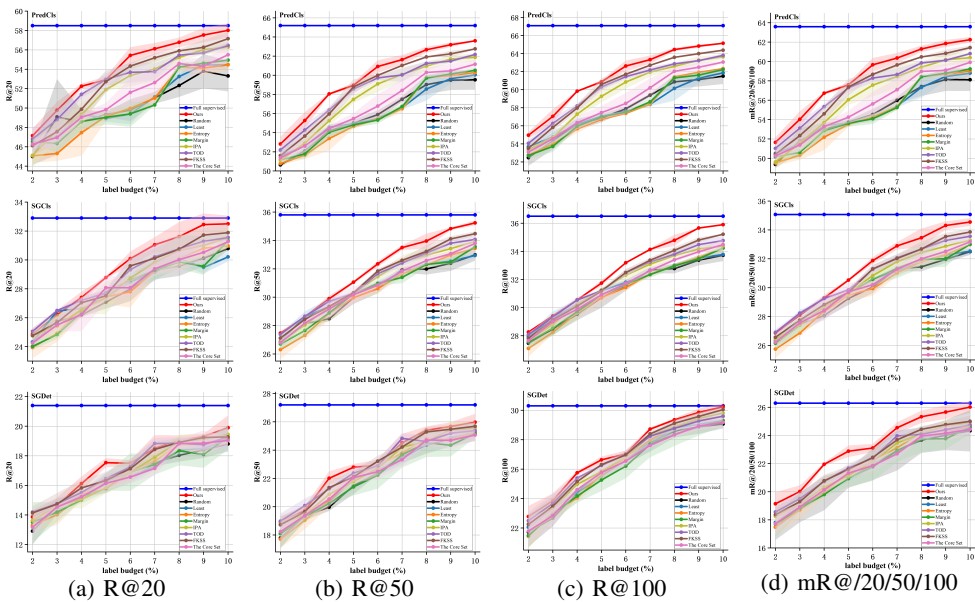

(a) R@20          (b) R@50          (c) R@100          (d) mR@/20/50/100

Figure 11: Quantitative evaluation on the VG150 dataset using MotifsNet (Zellers et al., 2018).

## B.4 DETAILED EXPERIMENTAL RESULTS

In this section, we provide more details on experimental results for benchmarking purposes with further analysis. In Figure 3, we have shown the average performance of R@/20/50/100 due to space limitations. Here we report more detailed quantitative results with different backbones of MotifsNet, VCTree, and Transformer in Figure 11, Figure 12, and Figure 13, respectively.

From these figures, we have the following observations and inspirations: **1)** In both challenging SGCls mode and SGDet evaluation modes, the performance of our proposed EDAL is closer to that of using full supervision than other methods. In addition, we see a great potential of active learning in SGG because all baseline methods achieve competitive performance given 10% of training labels, which further implies that discovering the most valuable data from long-tailed imbalanced SGG training data is worth further investigation. This gives us the confidence to continue the topic of this paper in the future. **2)** The overall trend of performance under R@50 and R@100 are very similar, and we suspect this is mainly caused by the lack of representativeness of the recall metric (R@K) with a larger value of K, especially when K is much higher than the actual number of relationships within the scene. To verify our assumption, we compute the histogram of the number of ground truth relationships and all possible relationships per image in the test set of VG150, shown in Figure 14. Where all possible relationships refer to the relationship between all object pairs. Specifically, $N$ objects have $N(N-1)$ possible relationships. We can see that the histogram of relationship statistics can support the above point. This observation shows that to accurately validate the SGG backbone, it is necessary to set an appropriate recall parameter according to the relationship distribution in the scene. **3)** Metric R@K is order-insensitive, *e.g.*, under R@50 there is no performance difference whether the successfully retrieved relation lies the 1-st or 50-th of the predictions. Moreover, R@K is K-dependent, *i.e.*, inappropriate K may affect the model evaluation. Thus, we aim to design a

K-independent and order-sensitive evaluation metric for SGG and which will be our pursuit in the immediate future.

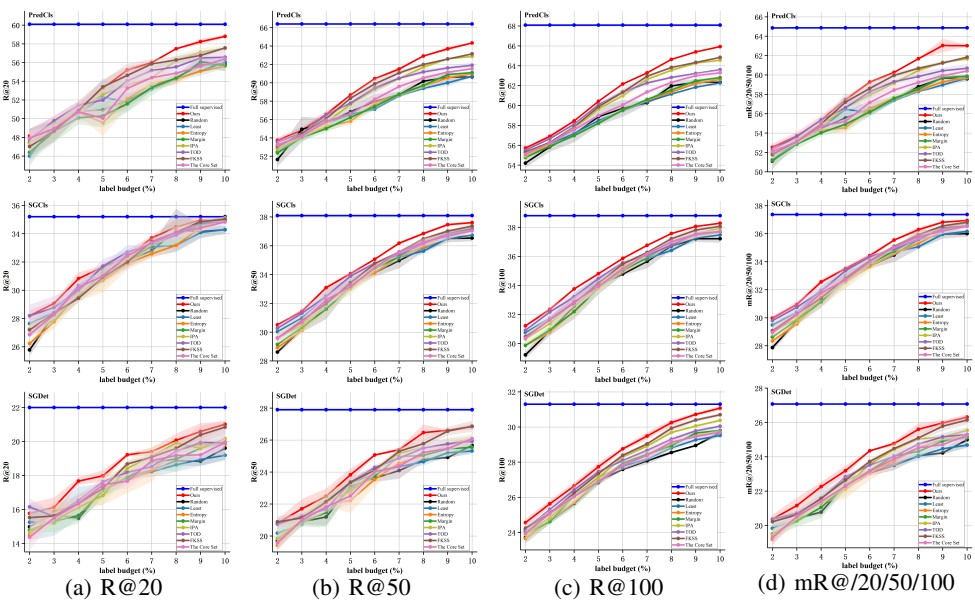

(a) R@20   (b) R@50   (c) R@100   (d) mR@/20/50/100

Figure 12: Quantitative evaluation on the VG150 dataset using VCTree (Tang et al., 2020b).

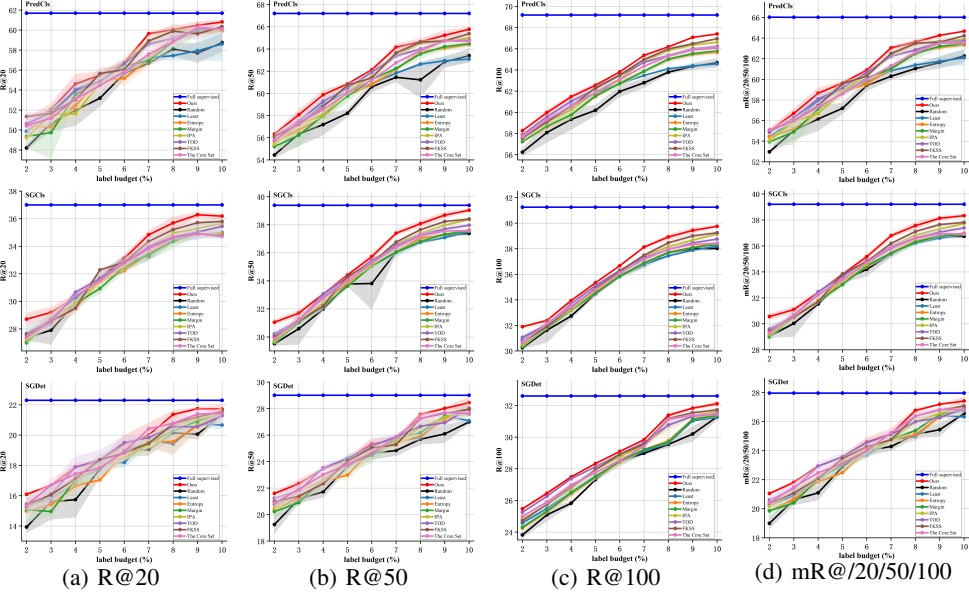

(a) R@20   (b) R@50   (c) R@100   (d) mR@/20/50/100

Figure 13: Quantitative evaluation on the VG150 dataset using Transformer (Shi & Tang, 2020).

## B.5 LABELING BUDGET UNDER EXPECTED MODEL PERFORMANCE

Figure 3 shows the evaluation results under the most commonly used metric for active learning, *i.e.*, performance under fixed labeling budget. In Table 4, Table 5, and Table 6, we report the performance under another active learning metric, *i.e.*, labeling budget under expected model performance. Under this metric, the less the labeling cost, the better the sampling of active learning approaches.

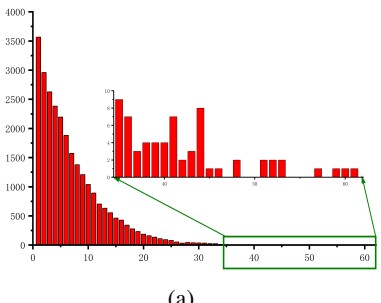 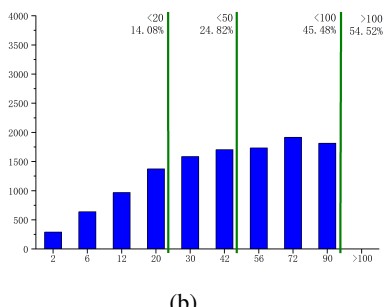

(a) (b)

Figure 14: The number distribution of ground truth relationships (a) and all possible relationships (b). The distribution shown here is counted come the test set of VG150.

From Table 4, Table 5, and Table 6, we can see that EDAL outperforms other baseline methods, which means that our proposed active learning framework requires less annotation cost when pursuing a fixed model performance. In addition, our experimental results also reveal another important advantage of EDAL: there will be a round gap between EDAL and the baseline methods. For example, when evaluated under the backbone of MotifsNet (Zellers et al., 2018), our proposed AL framework uses 7 rounds while the baselines take a minimum of 8 rounds to meet the expected performance (the expected performance here is that the R@20 reaches 30 in SGCls mode). The round gap shows that EDAL can reduce the labeling cost and also save training resources.

Table 4: Labeling budget under expected model performance. The SGG backbone used here is MotifsNet (Zellers et al., 2018).

| | Predicate classification (PredCls) | | | | | | Scene Graph Classification (SGCls) | | | | | | Scene Graph Detection (SGDet) | | | | | |
|---|---|---|---|---|---|---|---|---|---|---|---|---|---|---|---|---|---|---|
| | R@20 | | R@50 | | R@100 | | R@20 | | R@50 | | R@100 | | R@20 | | R@50 | | R@100 | |
| | 50 | 55 | 55 | 60 | 60 | 65 | 25 | 30 | 28 | 30 | 30 | 35 | 15 | 18 | 20 | 25 | 25 | 28 |
| random | 5.67 | 10.0 | 5.33 | 9.00 | 8.00 | 10.0 | 3.00 | 8.33 | 3.00 | 5.00 | 4.67 | 10.0 | 3.67 | 9.00 | 4.67 | 9.00 | 5.00 | 8.00 |
| Least | 6.00 | 10.0 | 5.67 | 9.67 | 8.33 | 10.0 | 3.00 | 9.00 | 3.00 | 4.67 | 4.33 | 10.0 | 4.33 | 8.33 | 4.00 | 10.0 | 5.00 | 7.33 |
| Entropy | 6.33 | 10.0 | 6.00 | 9.33 | 8.00 | 10.0 | 3.33 | 8.67 | 4.00 | 5.00 | 4.67 | 10.0 | 4.00 | 7.67 | 4.33 | 10.0 | 5.00 | 7.33 |
| Margin | 6.67 | 10 | 6.00 | 9.00 | 8.00 | 10.0 | 3.67 | 8.33 | 3.33 | 4.67 | 4.67 | 10.0 | 3.67 | 8.33 | 4.00 | 10.0 | 5.33 | 7.67 |
| IPA | 4.67 | 8.67 | 4.33 | 7.67 | 6.00 | 10.0 | 3.00 | 7.67 | 3.67 | 4.67 | 4.00 | 10.0 | 4.33 | 7.67 | 4.00 | 9.33 | 4.67 | 7.00 |
| TOD | 4.00 | 7.67 | 4.00 | 7.00 | 5.33 | 10.0 | 2.67 | 7.33 | 3.00 | 4.67 | 4.33 | 10.0 | 3.00 | 7.00 | 3.67 | 9.00 | 4.00 | 7.33 |
| FKSS | 4.67 | 7.33 | 4.33 | 6.33 | 5.33 | 10.0 | 2.67 | 7.67 | 3.00 | 4.67 | 4.33 | 10.0 | 3.67 | 6.67 | 3.67 | 8.33 | 4.33 | 7.00 |
| The Core Set | 5.67 | 8.67 | 5.33 | 8.67 | 7.00 | 10.0 | 3.00 | 8.67 | 3.00 | 4.67 | 4.67 | 10.0 | 4.33 | 7.67 | 4.00 | 9.00 | 5.00 | 8.33 |
| EDAL | 3.67 | 6.67 | 3.67 | 5.67 | 5.00 | 9.67 | 2.67 | 6.33 | 3.00 | 4.33 | 4.00 | 8.33 | 3.67 | 7.33 | 4.00 | 7.67 | 4.00 | 7.00 |

Table 5: Labeling budget under expected model performance. The SGG backbone used here is VCTree (Tang et al., 2020b).

| | Predicate classification (PredCls) | | | | | | Scene Graph Classification (SGCls) | | | | | | Scene Graph Detection (SGDet) | | | | | |
|---|---|---|---|---|---|---|---|---|---|---|---|---|---|---|---|---|---|---|
| | R@20 | | R@50 | | R@100 | | R@20 | | R@50 | | R@100 | | R@20 | | R@50 | | R@100 | |
| | 50.00 | 55.00 | 55.00 | 60.00 | 60.00 | 65.00 | 30.00 | 35.00 | 30.00 | 35.00 | 35.00 | 38.00 | 15.00 | 18.00 | 20.00 | 25.00 | 25.00 | 28.00 |
| random | 4.00 | 9.00 | 4.00 | 8.33 | 6.00 | 10.00 | 4.33 | 9.00 | 3.67 | 6.67 | 6.00 | 9.67 | 3.67 | 9.00 | 4.67 | 9.00 | 5.00 | 8.00 |
| Least | 4.33 | 9.33 | 4.00 | 9.67 | 7.00 | 10.00 | 4.00 | 8.67 | 2.33 | 6.33 | 5.67 | 10.00 | 4.33 | 8.33 | 4.00 | 10.00 | 5.00 | 7.33 |
| Entropy | 4.00 | 9.00 | 4.33 | 9.00 | 7.00 | 10.00 | 4.33 | 8.67 | 3.00 | 6.33 | 6.00 | 9.00 | 4.00 | 7.67 | 4.33 | 8.67 | 5.00 | 7.33 |
| Margin | 5.00 | 9.00 | 4.33 | 8.67 | 7.00 | 10.00 | 4.00 | 8.67 | 3.00 | 6.33 | 6.00 | 9.33 | 3.67 | 8.33 | 4.00 | 9.33 | 5.33 | 7.67 |
| IPA | 4.00 | 7.33 | 4.00 | 7.00 | 6.00 | 10.00 | 4.33 | 8.67 | 3.00 | 6.33 | 6.00 | 8.00 | 4.33 | 7.67 | 4.00 | 9.33 | 4.67 | 7.00 |
| TOD | 4.00 | 7.33 | 4.00 | 6.67 | 5.33 | 10.00 | 3.67 | 8.67 | 2.33 | 6.00 | 5.00 | 8.67 | 3.33 | 7.00 | 3.33 | 8.00 | 4.00 | 7.33 |
| FKSS | 4.33 | 6.33 | 4.00 | 6.33 | 5.67 | 10.00 | 4.33 | 8.00 | 2.33 | 6.00 | 5.67 | 8.00 | 3.67 | 6.67 | 3.67 | 8.33 | 4.33 | 7.00 |
| The Core Set | 4.67 | 8.33 | 4.00 | 7.67 | 6.67 | 10.00 | 3.67 | 8.33 | 3.00 | 6.00 | 6.00 | 8.67 | 4.33 | 8.00 | 4.00 | 9.00 | 5.00 | 8.00 |
| EDAL | 4.00 | 6.33 | 3.67 | 6.00 | 5.00 | 9.00 | 3.67 | 7.67 | 2.00 | 6.00 | 5.00 | 7.00 | 3.67 | 7.33 | 4.00 | 7.67 | 4.00 | 7.00 |

## B.6 Results with more label budget

In our main paper, we set the label budget to be 10%, *i.e.*, label 10% of the data in the unlabeled pool. In this section, we explore the performance of active learning under more label budgets. Specifically, we increase the budget to 13%, and the results are reported in Figure 15. From these experimental

results, we find that in Predicate classification (PredCls) mode, active learning can only obtain very trivial gains when increasing the budget. However, in Scene Graph Classification (SGCls) mode and Scene Graph Detection (SGDet) mode, active learning can continue to benefit from the increased budget and even over the full supervised performance. This promising result is exciting for the SGG task, which means that the active learning method can not only save considerable labeling costs but also further improve the model performance. We think this is mainly because the object information used by PredCls is ground-truth labels, while those used by SGCls and SGDet are obtained by a biased detector, *i.e.*, the detector is trained on data with the severe long-tailed distribution. In other words, both SGCls and SGDet try to predict the relationship between object pairs obtained by the biased detector. It is expected that the long-tailed distribution at the relationship-level further amplifies the detector bias. However, active learning has a certain mitigation effect on long-tailed distributions, which we believe is the reason why active learning methods can outperform the full supervised performance in some cases. In addition, two debiasing modules, CBM and IBM, are included in our active learning framework, which also plays a key role in improving model performance in SGCls and SGDet modes.

Further, we continue to increase the label budget to observe performance trends. Specifically, we increase the label budget to 20%, and the results are shown in Figure 16. The performance of EDAL begins to converge when the labeling budget reaches a certain value and increasing the labeling budget after that cannot improve performance. This inspires us that it is imperative to set a suitable stopping point for active learning.

Table 6: Labeling budget under expected model performance. The SGG backbone used here is Transformer-based backbone (Shi & Tang, 2020).

| | Predicate classification (PredCls) | | | | | | Scene Graph Classification (SGCls) | | | | | | Scene Graph Detection (SGDet) | | | | | |
|---|---|---|---|---|---|---|---|---|---|---|---|---|---|---|---|---|---|---|
| | R@20 | | R@50 | | R@100 | | R@20 | | R@50 | | R@100 | | R@20 | | R@50 | | R@100 | |
| | 50.00 | 55.00 | 60.00 | 65.00 | 60.00 | 65.00 | 30.00 | 33.00 | 30.00 | 35.00 | 35.00 | 38.00 | 15.00 | 20.00 | 22.00 | 25.00 | 25.00 | 30.00 |
| random | 3.33 | 6.00 | 6.33 | 10.00 | 5.00 | 10.00 | 3.33 | 6.00 | 3.00 | 7.33 | 7.33 | 7.44 | 3.33 | 8.67 | 4.67 | 7.33 | 3.67 | 9.67 |
| Least | 2.67 | 5.33 | 5.33 | 10.00 | 4.33 | 10.00 | 2.67 | 5.33 | 2.33 | 7.00 | 6.00 | 6.00 | 2.00 | 8.33 | 3.67 | 7.00 | 3.00 | 9.00 |
| Entropy | 2.33 | 6.67 | 6.00 | 10.00 | 4.67 | 8.33 | 3.33 | 6.33 | 3.33 | 7.00 | 6.67 | 6.89 | 2.33 | 8.33 | 4.00 | 6.67 | 3.00 | 9.00 |
| Margin | 3.00 | 5.67 | 5.67 | 10.00 | 4.67 | 8.33 | 4.00 | 6.00 | 3.33 | 7.00 | 6.67 | 6.89 | 2.67 | 7.67 | 4.00 | 6.67 | 3.00 | 9.00 |
| IPA | 3.00 | 5.67 | 5.33 | 10.00 | 4.33 | 7.67 | 3.67 | 6.00 | 3.33 | 7.00 | 6.00 | 6.00 | 2.67 | 7.67 | 4.00 | 7.33 | 3.00 | 8.33 |
| TOD | 2.00 | 5.33 | 5.00 | 10.00 | 4.00 | 8.00 | 2.67 | 5.67 | 2.00 | 7.00 | 6.00 | 6.00 | 2.33 | 7.00 | 3.33 | 6.33 | 2.00 | 8.00 |
| FKSS | 2.00 | 5.00 | 5.00 | 10.00 | 4.00 | 7.67 | 3.67 | 5.67 | 3.00 | 7.00 | 6.00 | 6.00 | 2.00 | 8.00 | 4.33 | 6.33 | 3.00 | 8.00 |
| The Core Set | 2.33 | 5.67 | 5.33 | 10.00 | 4.00 | 8.00 | 3.33 | 6.00 | 3.00 | 7.00 | 6.00 | 6.00 | 2.00 | 7.00 | 3.67 | 6.67 | 2.67 | 8.00 |
| EDAL | 2.33 | 5.00 | 4.67 | 9.00 | 3.33 | 7.00 | 2.67 | 5.00 | 2.00 | 6.00 | 6.00 | 6.00 | 2.00 | 7.00 | 3.33 | 6.33 | 2.00 | 8.00 |

## B.7 IMPLEMENTATION DETAILS OPEN-SET RELATIONSHIP RECOGNITION

VG150, a subset of the VG dataset, contains 150 object classes and 50 relationship classes. Therefore, as to V G150, there will be many open-set relationships in the VG dataset, *i.e.*, those labeled as foreground in VG but background in VG150. Our statistical result shows that there are about 68k open-set relationships in the test set of VG150. Similar to current work (Zellers et al., 2018; Tang et al., 2020a; Lu et al., 2016) evaluating on VG150 (Xu et al., 2017), to compare the effectiveness of different uncertainty estimation methods on *open-set* relationship recognition without contamination from data sampling, we conduct evaluations on the Visual Genome (VG) (Krishna et al., 2017) *test set*. Specifically, we first leverage the full training set of VG150 to train the model and use different strategies to estimate the relationship uncertainty. Then, we then classify the relationships based on the uncertainty, i.e. open-set relationships and non-open-set relationships, according to a fixed threshold (0.5 in our paper). We then classify the relationships based on the uncertainty, i.e. open-set relationships and non-open-set relationships, according to a fixed threshold (0.5 in our experiment). Finally, we rank the relationships for each scene according to uncertainty and compute the recall of open-set among the top-20/50/100 relationships. The performance on open-set relationship recognition is shown in Table 2.

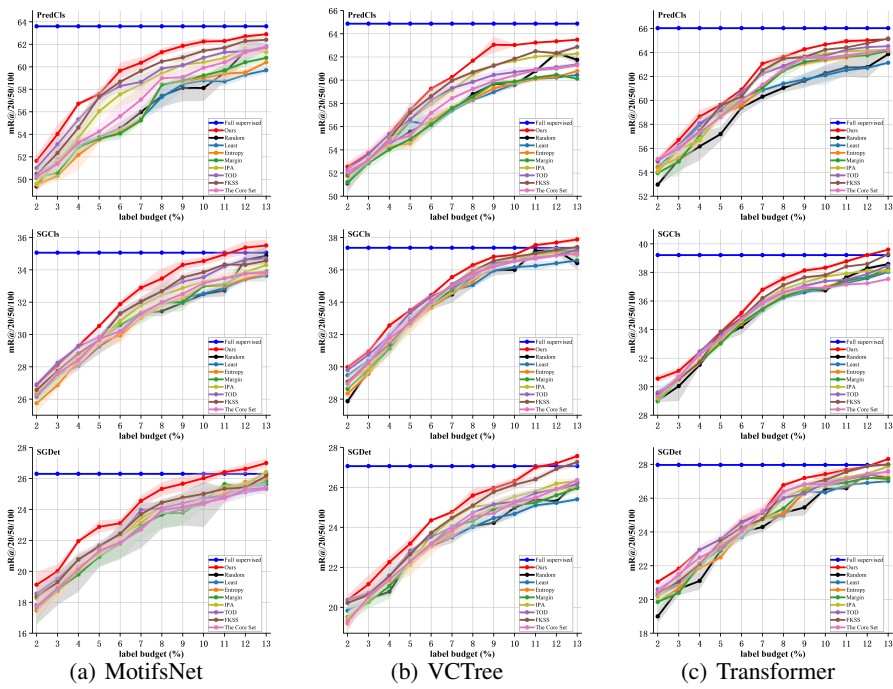

Figure 15: Active Learning performance of EDAL and baseline methods. mR@/20/50/100 represents the average performance of R@20, R@50, and R@100. We repeat each experiment three times and report the mean (solid line) and standard deviation (shadow). Unlike the label budget set (10%) in our main paper, we explore the model performance of more budgets here.

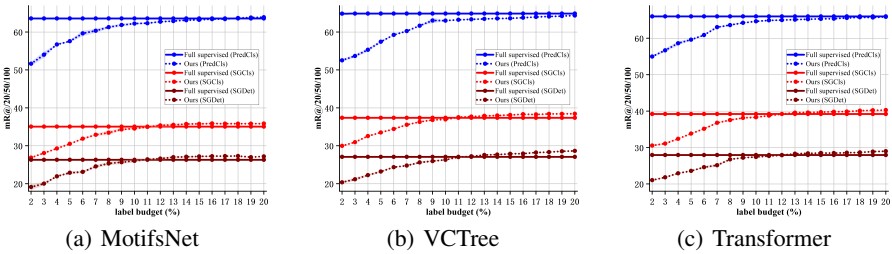

Figure 16: Active Learning performance of EDAL under 20% label budget.

## C APPENDIX TO CONCLUSION

The annotation expenses of SGG can be decoupled into relationship-level annotation and object-level annotations, of which the former is the most laborious. In addition, thanks to the existing large-scale datasets of object detection, our primary goal in this paper focus on reducing relationship annotations by leveraging the proposed active learning system EDAL. However, we found that the object-level annotations can be further decoupled into bounding box annotations and category annotations (see Figure 17). To further improve the practicability of active learning in SGG, it is worth more investigation to explore the joint active learning framework of both object-level and relationship-level annotations. We believe the correlation of multi-modal annotations would bring further benefits to improve the labeling efficiency of EDAL, which serves as our future work.

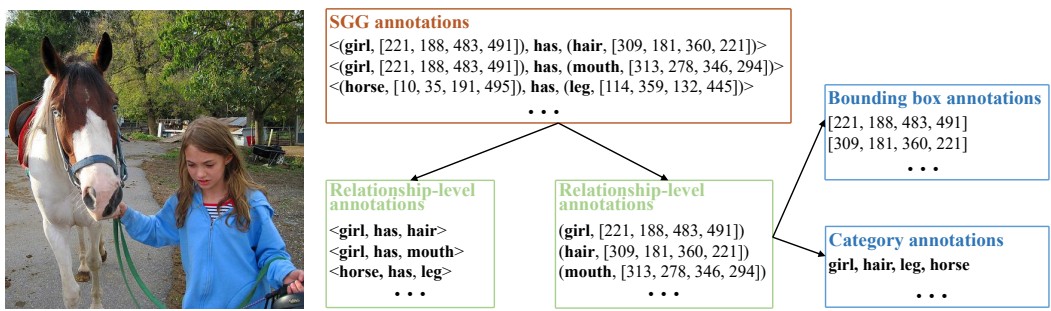

Figure 17: Decoupling of SGG annotations.

