# OpenReview forum: "Evidential Uncertainty and Diversity Guided Active Learning for Scene Graph Generation"
_ICLR.cc/2023/Conference — ICLR 2023 poster_

### Official Review · Reviewer_ffZd · 2022-10-22

**Confidence:** 4
**Correctness:** 4
**Technical Novelty And Significance:** 3
**Empirical Novelty And Significance:** 2
**Recommendation:** 6

**Clarity, Quality, Novelty And Reproducibility:**

The figure captions need to be more descriptive, such as Fig 2.
Minor things: Fig 4 title and subtitle aren't consistent. Not clear if the visualization outputs are from 3-rd or 5-th AL round of AL in the top-right section for each.

The main contribution is to aptly applying the EDL for uncertainty based AL and combining with diversity-based AL for SGG.

They have provided the code based on masked RCNN repo to substantiate reproducibility. Although I didn't try to run the code, but the structure of the code appears to be in accordance with the main contributions.


**Strength And Weaknesses:**

The paper aims to develop a hybrid approach utilizing both the uncertainty-based and diversity-based active learning for scene graph generation tasks. The paper utilizes evidential deep learning for its known property of being reliable specially in open-set problems.
They took inspiration from EDL originally aimed for open-set action recognition and modified it for relationship recognition. Essentially that is utilizing the evidential probability based cross entropy loss as a regularizer. This evidential probability is the second-order probability derived from the parameterized Dirichlet distribution. The hypothesis is that the cross-entropy loss based on evidential-probability prevents the overconfident predictions in open-set relationships without affecting the uncertainty estimates. This is an intended behavior specially because of sparse relationship triplet labels present in a scene. The experimental designs including baselines corresponding different sampling strategies are convincing. Fig 3 depicting mean Recall values with varying labeling budget is a valuable addition in this paper. It also shows that the improvement of mR values don't come at the cost of Recall values.

From the visualizations, it also looks like the method generates more fine-grained (in the case of VG, they are tail classes) relations : e.g. On vs sitting on in Fig 4. I am curious about how the relative performance trends are for a labeling budget over 10% such as 20%-25%.

**Summary Of The Paper:**

The paper proposes a method, EDAL, for scene graph generation with active learning. The paper combines an uncertainty-based and diversity-based active learning. The former utilizes evidential deep learning to estimate the relationship uncertainties and the diversity-based sampling is driven by image-level biases. The main idea is to utilize the evidential probability based cross entropy loss as a regularizer for relationship classification. This evidential probability is the second-order probability derived from the parameterized Dirichlet distribution. Additionally, diversity-based on context and image-level bias is incorporated.

**Summary Of The Review:**

The paper develops an hybrid active learning framework for learning SGG. The hybrid method utilizes both the diversity-based and EDL for uncertainty-based AL. One of the major contributions is modifying the regularization for the relationship recognition task using EDL. The results show their method archives higher mean Recall for PredCls, SGCls and SGDet, without compromising the performance of head classes as measured by Recall metrics within a relationship labeling budget of 10%.

---

> ### Author Response · Authors · 2022-11-13
> **Author Response to Reviewer ffZd**
>
> Thanks so much for your careful review and encouraging comments! We carefully address your concerns below.
>
> **Q1: From the visualizations, it also looks like the method generates more fine-grained (in the case of VG, they are tail classes) relations: e.g. On vs sitting on in Fig 4. I am curious about how the relative performance trends are for a labeling budget over 10\% such as 20\%-25\%.**
>
> Thank you for your nice suggestion. Following your suggestion, we have increased the annotation budget to 20\%. The results and analysis are added in Appendix B.6. For the performance trends, we have the following observations: 1) The performance of EDAL begins to converge when the labeling budget reaches a certain value, and increasing the labeling budget after that cannot improve performance. 2) The converged EDAL will approach or even exceed fully supervised performance. We argue that this mainly benefits from the fact that our method can alleviate the long-tailed distribution problem, which we analyze in detail in Appendix B.3.
>
> **Q2: The figure captions need to be more descriptive, such as Fig 2.**
>
> Thank you for giving such valuable suggestions. We have revised the captions of some figures (Figure 2 and Figure 4) to make them more clear.
>
> **Q3: Minor things: Fig 4 title and subtitle aren't consistent. Not clear if the visualization outputs are from 3-rd or 5-th AL round of AL in the top-right section for each.**
>
> Thanks for pointing us to this inconsistent caption. We have corrected it in the revision.

---

> ### Author Response · Authors · 2022-11-16
> **Further discussion with Reviewer ffZd**
>
> Dear Reviewer ffZd:
>
> Thank you again for the favorable assessment of our work. We gently remind you the reviewer-author discussion session started several days. We would appreciate it if you could have a chance to read our response as the discussion stage 1 is ending soon.
>
> According to your advice, we have examined the performance trends under the 20% labeling budget. Furthermore, we have carefully checked and revised the whole paper to ensure the paper is clear to read and understand.
>
> Please let us know whether you have any further concerns or suggestions to improve this work's quality.
>
> Thank you very much for your efforts and patience.
>
> Authors of paper 1782

---

> > ### Comment · Reviewer_ffZd · 2022-11-29
> > **20% labeling budget**
> >
> > Thank you for responding to my questions. I keep my original score, it'll be a nice paper to have.

---

> > > ### Author Response · Authors · 2022-11-30
> > > **Further discussion with Reviewer ffZd**
> > >
> > > Dear Reviewer ffZd:
> > >
> > > We appreciate your valuable comments that helped us in enhancing our paper. Thank you for your time and efforts for reviewing our paper.
> > >
> > > Do you have any other questions or suggestions? We are more than happy to answer any questions you may have.
> > >
> > > Authors of paper 1782

---

### Official Review · Reviewer_pmut · 2022-10-24

**Confidence:** 4
**Correctness:** 3
**Technical Novelty And Significance:** 3
**Empirical Novelty And Significance:** 2
**Recommendation:** 6

**Clarity, Quality, Novelty And Reproducibility:**

Overall, this article is well-organized and easy to follow. The authors have also provided the code in the supplementary material to ensure the reproducibility of the paper to some extent.

**Strength And Weaknesses:**

The positive aspects of the paper include:

1. Using the active learning method to save the annotation cost is a novel direction in scene graph generation and is important for further SGG study.

2. The designs of the proposed modules, i.e., EDL, RPG, CBM, and IBM, are convincing and fully consider the inherent problems of SGG, such as data bias and sparsity of the dataset.

3. The final performance of the proposed method is good, which approaches the performance of a fully supervised SGG model with only about 10% annotation cost for training the SGG model. The abundant ablation studies and detailed comparisons verified the effectiveness of the proposed modules.

There are some main concerns:

1. As stated in the abstract, the authors aim to reduce the annotation cost. However, the proposed method only reduces the training set for training models, and the method also needs a large labeled dataset to sample an informative training set. Does this approach really save annotation costs throughout data collecting and model training processes?

2. In section 3.1.1, the authors mentioned that “the uncertainty of xi can be estimated via evidential uncertainty ui = k/Si with a maximum value of 1.” Why is the maximum value 1? Is eij greater than or equal to 0? The definition of eij is unclear.

3. The authors should explain the qualitative results in Figure 4 in Section 4.2. What the figure illustrates?

Some minor issues:

1. Inconsistent symbol in Eq.(1) and following text. “S” (standardized) in Eq.(1) and “S” (italic) in text.

2. Inconsistent information in the caption and the image in Figure 4, “3-rd and 10-th AL rounds” in the caption and “5-th/10-th AL round” in the image.


**Summary Of The Paper:**

As a structured representation, the scene graph bridges computer vision and natural language processing, and the scene graph generation task is vital for the CV community. However, The annotation process of the scene graph is very complex including bounding boxes, labels, and relationships, so it is necessary to explore how to ensure the robustness of the scene graph model with few annotations. The authors propose a new method based on active learning, which trains a robust SGG model with fewer data by fully exploiting informative samples in the dataset.

**Summary Of The Review:**

Based on the above considerations, I believe that this paper has some novelty and will have some positive impact on the field. In the rebuttal stage, the authors should solve my main concerns to polish the final version.

---

> ### Author Response · Authors · 2022-11-13
> **Author Response to Reviewer pmut**
>
> Thanks for your insightful feedback! We appreciate your assessment about this paper being 'novelty' and 'positive impact'. We carefully address your concerns below.
>
> **Q1: As stated in the abstract, the authors aim to reduce the annotation cost. However, the proposed method only reduces the training set for training models, and the method also needs a large labeled dataset to sample an informative training set. Does this approach really save annotation costs throughout data collecting and model training processes?**
>
> Thank you for your nice question. There might be a little misunderstanding. Actually, to create an unlabeled dataset to evaluate our method, we first remove all relationship annotations in the training set of VG150. Therefore, our proposed framework, following the standard AL paradigm, samples an informative labeled training set from a large-scale unlabeled dataset. In this paper, our proposed AL framework only used 10\% relationship annotations in the training set of VG150.
>
> **Q2: In section 3.1.1, the authors mentioned that “the uncertainty of $x_i$ can be estimated via evidential uncertainty $u_i = k/S_i$ with a maximum value of 1.” Why is the maximum value 1? Is $e_{ij}$ greater than or equal to 0? The definition of $e_{ij}$ is unclear.**
>
> Our uncertainty calculation follows the evidential theory [1]. When all relationship categories share the same belief mass assignment, the evidential uncertainty achieves the maximum value of 1. For example, for input $x_i$, its output logits are all 0, i.e., $e_{ij}=0, j \in [0,k]$. Then, $S_i = \sum(e_{ij}+1)=k$, and thus $u_i = k/S_i=k/k=1$.
>
> $e_{ij}$ in our paper is the $j$-th output logit of $x_i$, and it acts as the collected evidence. Therefore, $e_{ij}$ can be greater than 0, less than 0, or equal to 0. Thanks for pointing us to this definition. We have redefined it to make it more clear.
>
> [1] Murat Sensoy, Lance Kaplan, and Melih Kandemir. Evidential deep learning to quantify classification uncertainty. Advances in neural information processing systems, 31, 2018.
>
>
> **Q3: The authors should explain the qualitative results in Figure 4 in Section 4.2. What the figure illustrates?**
>
> Thank you for your detailed advice! Figure 4 shows the results of our method at different AL rounds and the fully supervised model, highlighting that our AL framework can predict more fine-grained/meaningful relationships. Following your advice, we have added explanations of this figure in Section 4.2 and given more analysis in Appendix B.3.
>
> **Q4: Inconsistent symbol in Eq.(1) and following text. “S” (standardized) in Eq.(1) and “S” (italic) in text.**
>
> Thanks for pointing us this typo. We have corrected it in the revision.
>
> **Q5: Inconsistent information in the caption and the image in Figure 4, “3-rd and 10-th AL rounds” in the caption and “5-th/10-th AL round” in the image.**
>
> Thanks for pointing us to this inconsistent caption. We have corrected it in the revision.

---

> ### Author Response · Authors · 2022-11-16
> **Further discussion with Reviewer pmut**
>
> Dear Reviewer pmut:
>
> Thank you again for the favorable assessment of our work. We gently remind you the reviewer-author discussion session started several days. We would appreciate it if you could have a chance to read our response as the discussion stage 1 is ending soon.
>
> According to your advice, we have carefully checked and revised the whole paper, especially the typos and captions, to ensure the paper is clear to read and understand.
>
> Please let us know whether you have any further concerns or suggestions to improve this work's quality.
>
> Thank you very much for your efforts and patience.
>
> Authors of paper 1782

---

### Official Review · Reviewer_QrN3 · 2022-10-25

**Confidence:** 5
**Correctness:** 4
**Technical Novelty And Significance:** 3
**Empirical Novelty And Significance:** 3
**Recommendation:** 6

**Clarity, Quality, Novelty And Reproducibility:**

- Clarity: overall this article is well-organized and easy to follow.
- Quality and novelty: the paper is solid and it is an application-level novelty to apply the idea of active learning to the scene graph generation task.
- Reproducibility: The authors promise to release  the code.

**Strength And Weaknesses:**

#### Strengths
- Overall the paper is well written and easy to follow.
- It is interesting to see the authors apply active learning in the task of scene graph generation, which is not well explored so far.

#### Weaknesses
- Is the proposed method able to handle the label noise? Usually we cannot guarantee that the human annotations are 100% accurate and therefore a solid active learning method should also be able to handle the label noise to some extent.
- Are the orders of the uncertainty based module and the diversity based module exchangeable?
- Is the proposed task-specific active learning method able to outperform the task-agnostic active learning works like:

[1] Yoo, Donggeun, and In So Kweon. "Learning loss for active learning." In Proceedings of the IEEE/CVF conference on computer vision and pattern recognition, pp. 93-102. 2019.





**Summary Of The Paper:**

The paper proposes an evidential uncertainty and diversity guided deep active learning framework for the scene graph generation task. In particular, uncertainty is estimated by coupling evidential deep learning (EDL) and global relationship mining. Also, a context blocking module (CBM) and image blocking module (IBM) are designed to explore the diversity existing with context-level and image-level bias. The experiments on the VG150 dataset seem to validate the effectiveness of the proposed method.

**Summary Of The Review:**

Based on the above statements, I would like to weakly accept the paper at the initial stage. After carefully reading the authors' response, I am happy to see most of my concerns have been well addressed and therefore would like to keep the initial rating.

---

> ### Author Response · Authors · 2022-11-13
> **Author Response to Reviewer QrN3 (Page 1)**
>
> Thank you for your constructive feedback and acknowledgements! We appreciate your assessment about this paper being `solid'. Below we carefully address your concerns.
>
> **Q1: Is the proposed method able to handle the label noise? Usually we cannot guarantee that the human annotations are 100\% accurate and therefore a solid active learning method should also be able to handle the label noise to some extent.**
>
> Thank you for asking this question. Our proposed method is robust to certain human annotation noises, considering that annotating the SGG dataset is very costly and comes with a lot of noises, such as missing relationships, wrong relationships, fine-grained relationships labeled as coarse-grained ones, etc., which have been observed in many literature as well. For missing relationships, our method estimates the uncertainty of (background/foreground) relationships that exist between all object pairs. As to wrong relationships, our proposed EDL-based method has been proved to be work robustly with OOD data (Table 2 of the main paper). For fine-grained relationships labeled as coarse-grained ones, our proposed method pays more attention to the tail categories and thus predicts fine-grained relationships in most cases. We have added related discussions about this advantage in Section 4.2 and Appendix B.6.
>
> **Q2: Are the orders of the uncertainty based module and the diversity based module exchangeable?**
>
> The uncertainty based module and the diversity based module are exchangeable. Inspired by your comment, we changed the order of the above two modules, and the result on MotifsNet backbone is as follows:
>
> **Table:  Performance on open-set relationship recognition. The SGG backbone used here is MotifsNet**
> | order | PredCls           | SGCls             | SGDet             |
> |------------------------------------------------------------------|-------------------|-------------------|-------------------|
> |                                                                  | R@20/50/100       | R@20/50/100       | R@20/50/100       |
> | Diversity $\\rightarrow$ Uncertainty                             | 55\.1/61\.7/62\.6 | 28\.9/31\.9/29\.8 | 18\.4/25\.3/28\.7 |
> | Uncertainty $\\rightarrow$ Diversity                             | 57\.0/63\.5/65\.1 | 31\.7/35\.0/35\.8 | 21\.4/27\.2/30\.3 |
>
> From the above results, we can find that the reorder-based framework performs poorly on the SGG task. We argue that this is due to the fact that the SGG task is dependent on the complex features of object categories, object boxes, and relationships between objects, thus, it is difficult to obtain the representative samples we expect using the diversity-based module first. Furthermore, using the diversity-based module to reduce the unlabeled data pool to a small range also greatly limits the contribution of the uncertainty estimation module. Finally, the reorder-based framework requires more computational overhead since it performs density matching on the full unlabeled pool. In contrast, the uncertainty-based module in our framework only needs to compute the density distance within a subset (1.2\% of the unlabelled pool in our paper).

---

> > ### Author Response · Authors · 2022-11-13
> > **Author Response to Reviewer QrN3 (Page 2)**
> >
> > **Q3: Is the proposed task-specific active learning method able to outperform the task-agnostic active learning works like: [1] Yoo, Donggeun, and In So Kweon, ``Learning loss for active learning.'' CVPR, 2019.**
> >
> > Thanks for pointing out this related work. We have added in the revision the comparison with ``Learning loss for active learning (Loss Prediction Module, LPM) [1]'', with the results shown in Table 2. LPM devises a novel uncertainty-based sampling strategy: the Loss Prediction Module estimates uncertainty based on the inference loss of unlabeled input data. From Table 2, the experimental results show that LPM could outperform some uncertainty-based methods such as LCS and TOD. However, LPM is weaker than methods such as FKSS and our proposed framework. It might be that LPM co-supervises the SGG model with the cross-entropy loss, so overconfident predictions on open-set relationships cannot be avoided. In contrast, our proposed EDL-based module can get softer predictions by placing a Dirichlet distribution over the class probabilities.
> >
> > **Table 2: Performance on open-set relationship recognition. The SGG backbone used here is MotifsNet.**
> > |                                            |                                 |                                 |                                 |
> > |-------------------------------------------------|---------------------------------|---------------------------------|---------------------------------|
> > | **Method** | **PredCls**    | **SGCls**       | **SGDet**      |
> > |                                                 | R@20/50/100 | R@20/50/100 | R@20/50/100 |
> > | EDAL+LCS                                        | 50.1/62.4/70.1                  | 34.3/40.4/43.7                  | 14.4/20.4/25.5                  |
> > | EDAL+TOD                                        | 53.2/66.0/73.8                  | 34.0/42.2/44.3                  | 15.8/22.0/27.0                  |
> > | EDAL+FKSS                                       | 57.3/69.4/76.8                  | 35.2/43.1/46.2                  | 17.1/23.4/28.8                  |
> > | EDAL+LPM                                        | 55.2/68.4/75.6                  | 34.7/42.6/45.3                  | 16.3/22.7/28.1                  |
> > | EDAL                                            | 59.7/72.7/80.6                  | 37.1/44.6/48.4                  | 17.7/25.2/30.3                  |
> >
> > [1] Donggeun Yoo and In So Kweon. Learning loss for active learning. In Proceedings of the IEEE conference on computer vision and pattern recognition, pp. 93–102, 2019.

---

> ### Author Response · Authors · 2022-11-16
> **Further discussion with Reviewer QrN3**
>
> Dear Reviewer QrN3:
>
> Thank you again for the favorable assessment of our work. We gently remind you the reviewer-author discussion session started several days. We would appreciate it if you could have a chance to read our response as the discussion stage 1 is ending soon.
>
> Based on your suggestions, we have added some key experiments, including exchanging the uncertainty based module with the diversity based module, and the comparison of task-agnostic active learning works.
>
> Please let us know whether you have any further concerns or suggestions to improve this work's quality.
>
> Thank you very much for your efforts and patience.
>
> Authors of paper 1782

---

> > ### Author Response · Authors · 2022-11-30
> > **Further discussion with Reviewer QrN3**
> >
> > Dear Reviewer QrN3:
> >
> > We highly appreciate your efforts in reviewing our paper. We have revised our paper according to the reviews' comments, added some experiments (e.g., increased label budget to 20%), further enhanced the text, smoothed the presentation, and proofread the revised paper to ensure that there are no typos. Revised contents are colored in blue in the revision.
> >
> > Thank you again for your time and efforts! Please let us know if you have any concerns or suggestions. We look forward to discussing more with you.
> >
> > Authors of paper 1782

---

### Official Review · Reviewer_Jsof · 2022-11-03

**Confidence:** 4
**Correctness:** 2
**Technical Novelty And Significance:** 2
**Empirical Novelty And Significance:** 1
**Recommendation:** 5

**Clarity, Quality, Novelty And Reproducibility:**

-Clarity: The writing could be improved to make this paper more clear. -Quality: The overall of this paper is good. -Novelty: The novelty is incremental and limited. -Reproducibility: It seems to be reproducible but some details are missing.

**Details Of Ethics Concerns:**

None.

**Strength And Weaknesses:**

Strength:
(1) Adopting evidential uncertainty and diversity guided deep active learning can alleviate the issue of open-set relationships sounds.
(2) The proposed method can save human annotation costs by approaching the performance of the fully supervised model seems promising.


Weakness:
(1) How to evaluate the robustness of proposed uncertainty method? Please offer more details.
(2) How to handle the long-tail problem with the proposed model? Please examine the effect of the method.

**Summary Of The Paper:**

The paper proposed an active learning-base model for scene graph generation, in order to reduce the prohibitively expensive annotation cost.

**Summary Of The Review:**

Regarding the limited contribution and unclear details of this paper, I recommed to weakly reject the paper.

---

> ### Author Response · Authors · 2022-11-13
> **Author Response to Reviewer Jsof**
>
> Thank you for your insightful comments and helpful suggestions to improve the paper! We thank the reviewer for suggesting adding constructive evaluation and examination. We carefully address your concerns below.
>
> **Q1: How to evaluate the robustness of proposed uncertainty method? Please offer more details.**
>
> Thank you for asking this question. We do evaluate the robustness of the proposed uncertainty method by recognizing the open-set relationships (Table 2 of the main paper). Our evaluation procedure is inspired by [1], which emphasizes that a robust uncertainty method should perform well for out-of-distribution (OOD) data. However, existing uncertainty methods, especially those based on cross-entropy loss, perform poorly on open-set relationship recognition due to their false over-confidence predictions. In contrast, our proposed EDL-based uncertainty method can obtain robust uncertainty estimates by collecting evidence from each output class by placing a Dirichlet distribution over the class probabilities. We show in Table 2 that our proposed uncertainty method outperforms baseline uncertainty methods by a clear margin. Due to space constraints, we provide more details in Appendix B.7. Please refer to this subsection.
>
> **Table 2: Performance on open-set relationship recognition. The SGG backbone used here is MotifsNet.**
> |                                            |                                 |                                 |                                 |
> |-------------------------------------------------|---------------------------------|---------------------------------|---------------------------------|
> | **Method** | **PredCls**    | **SGCls**       | **SGDet**      |
> |                                                 | R@20/50/100 | R@20/50/100 | R@20/50/100 |
> | EDAL+LCS                                        | 50.1/62.4/70.1                  | 34.3/40.4/43.7                  | 14.4/20.4/25.5                  |
> | EDAL+TOD                                        | 53.2/66.0/73.8                  | 34.0/42.2/44.3                  | 15.8/22.0/27.0                  |
> | EDAL+FKSS                                       | 57.3/69.4/76.8                  | 35.2/43.1/46.2                  | 17.1/23.4/28.8                  |
> | EDAL+LPM                                        | 55.2/68.4/75.6                  | 34.7/42.6/45.3                  | 16.3/22.7/28.1                  |
> | EDAL                                            | 59.7/72.7/80.6                  | 37.1/44.6/48.4                  | 17.7/25.2/30.3                  |
>
> [1] Anna-Kathrin Kopetzki, Bertrand Charpentier, Daniel Zügner, Sandhya Giri, and Stephan Günnemann. Evaluating robustness of predictive uncertainty estimation: Are dirichlet-based models reliable? In International Conference on Machine Learning, pp. 5707–5718. PMLR, 2021.
>
> **Q2: How to handle the long-tail problem with the proposed model? Please examine the effect of the method.**
>
> Thank you for your valuable suggestions. Our method handles the long-tail problem by paying more attention to tail relationships. Our proposed EDL-based method is able to predict high uncertainty estimates for tail category relationships even with the head-dominant model. Following your advice, we have examined the effect of our method on handling the long-tail problem in Appendix B.3. Specifically, we report the sampling distribution of our method at different rounds of the active learning process, as well as the complete training data distribution. Results show that the proportion of head categories in the full data is 78\%, but is less than 50\% in our sampled training data. Please see Appendix B.3 for more results and analysis.

---

> ### Author Response · Authors · 2022-11-16
> **Further discussion with Reviewer Jsof**
>
> Dear Reviewer Jsof:
>
> Thank you again for the constructive assessment of our work. We gently remind you the reviewer-author discussion session started several days. We would appreciate it if you could have a chance to read our response as the discussion stage 1 is ending soon.
>
> We have provided more details on how to evaluate the robustness of our proposed uncertainty method in response to your nice advice. Furthermore, we have analyzed why our method can handle the long-tail problem as well as examined the effect.
>
> Please let us know whether you have any further concerns or suggestions to improve this work's quality.
>
> Thank you very much for your efforts and patience.
>
> Authors of paper 1782

---

> > ### Author Response · Authors · 2022-11-30
> > **Further discussion with Reviewer Jsof**
> >
> > Dear Reviewer Jsof:
> >
> > We highly appreciate your efforts in reviewing our paper. We have revised our paper according to the reviews' comments, added some experiments (e.g., increased label budget to 20%), further enhanced the text, smoothed the presentation, and proofread the revised paper to ensure that there are no typos. Revised contents are colored in blue in the revision.
> >
> > Thank you again for your time and efforts! Please let us know if you have any concerns or suggestions. We look forward to discussing more with you.
> >
> > Authors of paper 1782

---

### Author Response · Authors · 2022-11-13
**General response to Reviewers**

We appreciate the valuable comments from the reviewers that helped us in enhancing our paper. The whole paper has been revised according to each revision suggestion and the revised paper has been proofread to ensure the paper is easy to read, understand, and as clear as possible regarding the description and the evaluation of the proposed method. The changes in the new revision of our draft are marked in blue.

In summary, we address the following concerns:

$\bullet$ We extended the discussion to show that our proposed uncertainty method is robust and clarified how our proposed AL framework alleviates the long-tailed distribution problem.

$\bullet$ We explored the exchangeability of the uncertainty based module and the diversity based module and added the comparison with a novel uncertainty-based sampling strategy.

$\bullet$ We added detailed explanations of Figure 4 and updated some typos to make them more clear.

$\bullet$ We added experiments to observe the trend of our proposed AL framework under more labeling budgets.

---

### Decision · Program_Chairs · 2023-01-20

**Decision:**

Accept: poster

**Justification For Why Not Higher Score:**

Considering the overall contribution of this work and the quality of this paper, this paper is marginally above the acceptance threshold.

**Justification For Why Not Lower Score:**

This paper studies how to leverage active learning for scene graph generation, which would inspire researchers in this field.

**Metareview: Summary, Strengths And Weaknesses:**

This paper presents an evidential uncertainty and diversity guided deep active learning framework for scene graph generation. The key idea of this work is to utilize the evidential probability based cross entropy loss as a regularizer for relationship classification. Experimental results on the VG150 dataset are provided and discussed.

Overall, this paper is well organized and clearly written. Using active learning for scene graph generation is an interesting direction. The motivation of the proposed method is clear, and experiments demonstrate its effectiveness.

Meanwhile, reviewers raised some concerns such as technical details, experimental settings, and analysis of the qualitative analysis. The authors provided detailed responses and additional experimental results, which can well address most of the concerns from reviewers. The authors are strongly encouraged to incorporate the suggestions from reviewers as well as the new results to the final version.

**Note From Pc:**

if the above contains the word "oral" or "spotlight" please see: "oral" presentation means -> notable-top-5% and "spotlight" means -> notable-top-25%. As stated in our emails, we are disassociating presentation type from AC recommendations

**Summary Of Ac-Reviewer Meeting:**

N/A